

# AlphaViT: a flexible game-playing AI for multiple games and variable board sizes

Kazuhisa Fujita

Department of Clinical Engineering, Komatsu University, Komatsu, Japan

## ABSTRACT

This study presents three game-playing agents incorporating Vision Transformers (ViTs) into the AlphaZero framework: AlphaViT, AlphaViD (AlphaViT with a transformer decoder), and AlphaVDA (AlphaViD with learnable action embeddings). These agents can play multiple board games with varying sizes using a single shared-weight neural network, thus overcoming AlphaZero's limitation of fixed board sizes. AlphaViT employs only a transformer encoder, whereas AlphaViD and AlphaVDA incorporate both a transformer encoder and a decoder. In AlphaViD, the decoder processes the output from the encoder, whereas AlphaVDA uses learnable embeddings as decoder input. The additional decoder in AlphaViD and AlphaVDA provides flexibility to adapt to various action spaces and board sizes. Experimental results show that the proposed agents, trained on either individual games or multiple games simultaneously, consistently outperform traditional algorithms, such as Minimax and Monte Carlo Tree Search. They achieve performance close to that of AlphaZero despite relying on a single deep neural network (DNN) with shared weights. In particular, AlphaViT performs well across all evaluated games. Furthermore, fine-tuning the DNN using weights pre-trained on small board games accelerates convergence and improves performance, particularly in Gomoku. Notably, simultaneous training on multiple games yields performance comparable to, or even surpassing, that of single-game training. These results indicate the potential of transformer-based architectures for developing flexible and robust game-playing AI agents that excel in multiple games and dynamic environments.

## INTRODUCTION

Artificial intelligence (AI) has advanced remarkably in recent years, demonstrating its potential across a wide range of applications. One area where AI has excelled is mastering board games, often outperforming top human players. Notable achievements include AI agents that have defeated humans in games such as Checkers (*Schaeffer et al., 1993*), Chess (*Campbell, Hoane & hsiung Hsu, 2002*), and Othello (*Buro, 1997*). A significant turning point occurred in 2016 when AlphaGo (*Silver et al., 2016*), an AI designed specifically for the game of Go, defeated the world's top players. Subsequently, AlphaZero (*Silver et al., 2018*) was introduced, demonstrating its capability to master various board games, including Chess, Shogi, and Go. These achievements further highlight AI's superhuman skill in this domain.

Corresponding author
Kazuhisa Fujita,
kazu@spikingneuron.net

Despite these successes, many current game-playing AI agents suffer from a fundamental limitation: they are typically designed for just one specific game and cannot play other games. Even within the same game, these agents cannot handle variations in board size. In contrast, humans can easily switch between different board sizes. For example, beginners in Go often start practicing on smaller boards (*e.g.*, 9 × 9) before progressing to larger boards (*e.g.*, 19 × 19). However, AI agents like AlphaZero, which are designed for a single specific game and a fixed board size, require substantial reprogramming to accommodate such changes.

For AlphaZero, this limitation arises from its deep neural network (DNN) architecture, which requires a fixed input size. AlphaZero's DNN consists of residual blocks and multilayer perceptrons (MLPs) designed for a fixed input size. The output size of the residual blocks varies with changes in input size, creating inconsistencies with the expected MLP input size. Consequently, AlphaZero fails even with small changes in board size.

To address this limitation, this study proposes replacing residual blocks in the AlphaZero framework with Vision Transformer (ViT) (*Dosovitskiy et al., 2021*). ViT is an image-classification DNN based on the transformer architecture. ViT divides an image into patches, encodes them using a transformer, and classifies the image based on these encoded patches. A key advantage of ViT is its flexibility in handling various image sizes. This flexibility enables the AlphaZero framework to adapt to various games and board sizes.

This article presents game-playing agents capable of handling multiple games and variable board sizes using a single DNN. These agents, named AlphaViT, AlphaViD (AlphaViT with a transformer decoder), and AlphaVDA (AlphaViD with learnable action embeddings), are based on the AlphaZero framework. The agents predict the value of a game state and the policy using a DNN, and choose moves *via* Monte Carlo Tree Search (MCTS) (*Browne et al., 2012*; *Winands, 2017*). Computational experiments show that the proposed agents can be trained to play three games (Connect 4, Gomoku, and Othello) simultaneously using a single DNN with shared weights. Moreover, the proposed agents outperform traditional algorithms, such as Minimax and MCTS, across various games, while approaching the performance of AlphaZero, whether trained on a single game or multiple games simultaneously. The goal of this study is not to surpass the state-of-the-art specialized single-game agents, but rather to demonstrate that a single transformer architecture can flexibly handle multiple games and board sizes.

Portions of this text were previously published as part of a preprint (https://arxiv.org/abs/2408.13871).

## RELATED WORK

Game-playing AI agents have reached superhuman performance levels in traditional board games such as Checkers (*Schaeffer et al., 1993*), Othello (*Buro, 1997*, *2003*), and Chess (*Campbell, 1999*; *Hsu, 1999*; *Campbell, Hoane & hsiung Hsu, 2002*). In 2016, AlphaGo (*Silver et al., 2016*), a Go-playing AI, defeated the world's top Go players, marking the first superhuman-level performance in Go. AlphaGo relied on supervised learning from a large

database of expert human moves and self-play data. Subsequently, AlphaGo Zero (*Silver et al., 2017*) defeated AlphaGo, relying only on self-play data. In 2018, *Silver et al. (2018)* proposed AlphaZero, which has no restrictions on the types of games it can play. AlphaZero outperformed other superhuman-level AIs in Go, Shogi, and Chess. Interestingly, AlphaZero's capabilities extend beyond traditional two-player perfect-information games, with research exploring its potential in more complex scenarios. For example, *Hsueh et al. (2018)* showed AlphaZero's potential in nondeterministic games. Other extensions include handling continuous action spaces (*Moerland et al., 2018*) and supporting multiplayer games (*Petosa & Balch, 2019*). However, a practical limitation of AlphaZero stems from its DNN design, as its policy and value heads include MLP layers that require fixed input shapes. This ties a trained network to a single board size and game, preventing one shared-weight model from handling multiple games or size variants.

Researchers have improved the AlphaZero framework to overcome its limitations. *Wu (2019)* tackled some of these limitations by improving the efficiency of AlphaZero-like training in Go. Wu's model introduced techniques such as *playout cap randomization* and *policy target pruning*, which significantly accelerate self-play learning. Wu used global pooling layers to standardize varying board sizes to a fixed size, enabling the model to estimate values using MLPs effectively. Furthermore, Wu exclusively utilized convolutional layers for policy estimation, thereby eliminating the need for MLPs. This design enables seamless handling of varying board sizes while efficiently calculating the policy. This innovation represents a significant step toward creating more flexible AI agents capable of adapting to different board configurations. Similarly, *Soemers et al. (2023)* explored transfer learning across various board games, employing fully convolutional networks with global pooling to enable effective transfer between games with different board sizes, shapes, and rules. Their approach has demonstrated the potential of convolutional architectures, enhanced with global pooling, to generalize across various game scenarios.

To address these limitations, this study proposes integrating a transformer architecture into the AlphaZero framework. A transformer architecture, initially developed for natural language processing (*Vaswani et al., 2017*), has proven remarkably effective across domains, including image-processing tasks. Transformer-based models achieve exceptional performance in various image-related tasks, such as image classification (*Dosovitskiy et al., 2021*), semantic segmentation (*Xie et al., 2021*), video classification (*Li et al., 2022*), and video captioning (*Zhao et al., 2022*). ViT, introduced by *Dosovitskiy et al. (2021)*, is a notable example of a transformer-based model for image processing. ViT achieved state-of-the-art performance in image classification at the time of its introduction. A key feature of ViT is its independence from the input image size (*Dosovitskiy et al., 2021*). Unlike convolutional neural networks, which require fixed-size inputs, ViT can process images of various sizes. It achieves this by dividing each image into fixed-size patches, each treated as a token in the transformer architecture. This flexibility makes ViT highly adaptable and efficient when handling different image sizes. By incorporating the ViT architecture into the AlphaZero framework, this study aims to

extend its capabilities to handle various games with different board sizes and enhance its flexibility.

Some researchers have explored the use of transformers in game-playing AI. For example, *Czech et al. (2024)* proposed a variant of the AlphaZero framework, AlphaVile, using a novel network architecture with modified lightweight transformer blocks and an original loss function. AlphaVile achieved better performance than AlphaZero in Chess. *Ruoss et al. (2024)* reached grandmaster level in Chess with a transformer model trained solely on a dataset, without any self-play or game-tree search. *Monroe & Chalmers (2024)* proposed a transformer-based architecture for Chess-playing AI. Using a simple transformer architecture, they trained agents that reached a high level of play. These single-game successes suggest that a transformer architecture can replace convolutions, and this study takes the next step and shows that a transformer can simultaneously master multiple games and board sizes with a single set of weights.

## METHODS: ALPHAVIT, ALPHAVID, AND ALPHAVDA

AlphaZero's game-playing capability is limited to games with specific board sizes and rules used during training, as discussed in previous sections. This limitation arises from AlphaZero's DNN architecture, in which MLPs require fixed input sizes. To address this limitation, this study proposes AlphaViT, AlphaViD, and AlphaVDA as game-playing AI agents based on the AlphaZero framework but using ViT architecture. These agents use a combination of a DNN and MCTS (Fig. 1). The DNN receives the board state and outputs a value estimate and move probabilities (policies). The MCTS searches a game tree using the estimated value and move probabilities. By incorporating ViT instead of residual blocks, AlphaViT, AlphaViD, and AlphaVDA can overcome the limitations of AlphaZero and play games that have variable board sizes and rule sets. While AlphaViT employs only a transformer encoder, AlphaViD and AlphaVDA employ both a transformer encoder and a decoder. Importantly, AlphaViT, AlphaViD, and AlphaVDA can play any game that AlphaZero can because they employ the same game-playing algorithm as AlphaZero (see 'AlphaZero' for details).

The training method for AlphaViT, AlphaViD, and AlphaVDA is identical to that of AlphaZero, consisting of three stages: self-play, augmentation, and update. In the self-play stage, the agent generates training data by playing games against itself. The augmentation stage applies data augmentation techniques to the training data. Finally, during the update stage, the DNN weights are updated using the augmented training data. The details of the training method are described in 'Training procedure'.

### Architectures of DNNs

**AlphaViT** An overview of AlphaViT's DNN architecture is shown in Fig. 2. The DNN in AlphaViT is based on ViT, which has no input-size limitation and can classify images even if the input image size differs from the training image size. This flexibility enables AlphaViT to play games with different board sizes using the same network. In AlphaViT, the game boards are fed into ViT. Initially, these inputs are transformed into patch embeddings through a convolutional layer. Using a convolutional layer allows easy

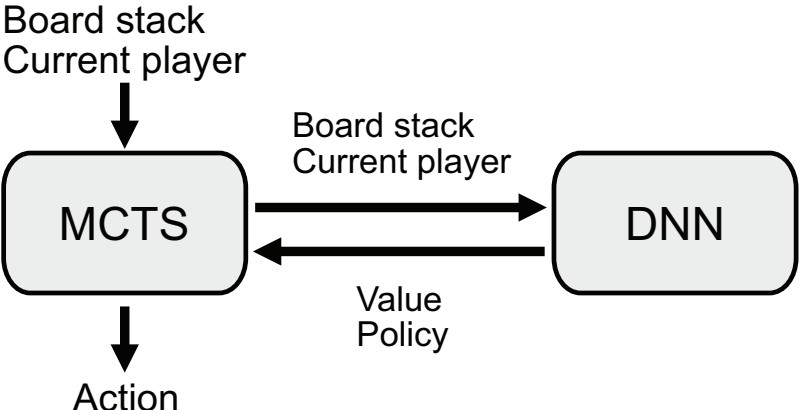

**Figure 1 Overview of the decision process.** The agents take a stack of board planes and the current player as input and determine the next move using Monte Carlo Tree Search (MCTS). MCTS explores the game tree using the value and policy provided by the DNN.

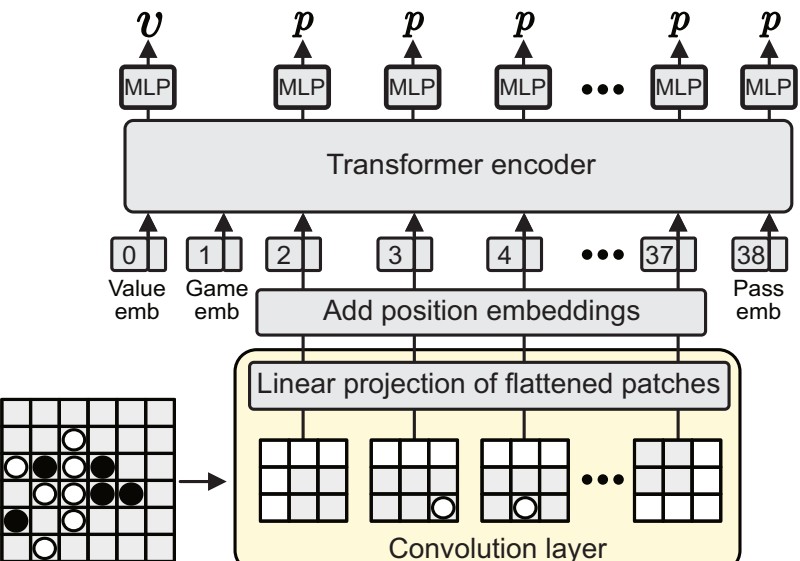

**Figure 2 AlphaViT architecture.** The input board is divided into patches by a convolutional layer, projected to patch embeddings, and combined with value, game, and pass tokens (embeddings). After adding positional embeddings, the sequence is processed by a transformer encoder. Outputs are used by MLP heads to predict the state value and move probabilities.

adjustment of the patch partitioning parameters, such as patch size, stride, and padding. The final output of the encoder is used to compute the value and move probabilities. AlphaViT employs three special tokens to enable flexibility for different games and board sizes:

- **Value token $\mathbf{x}_{\text{value}}$**: is fed to an MLP value head to predict the state value $v(s) \in (-1, 1)$, where $s$ is the game state, 1 indicates a certain win, and $-1$ a certain loss. This token is learnable.

- **Game token $\mathbf{x}_{\text{game}}$**: encodes which game is currently being played. It is represented as a non-trainable one-hot vector whose length is equal to the embedding size $D$. Each game type (*e.g.*, Connect 4, Gomoku, Othello) is assigned a unique index in this vector. Although the token itself is fixed, the model can learn game-specific representations through downstream layers. This design also enables flexible addition of new games, up to $D$ in number.

- **Pass token $\mathbf{x}_{\text{pass}}$**: represents the "pass" action that exists in Othello but not in Gomoku or Connect 4. Appending a dedicated learnable token vector after all patch embeddings, this enables the model to predict the probability of the pass move when the rules require it. This token is learnable.

Given a game state $s$, AlphaViT's DNN predicts the value $v(s)$ and the move probability vector $\boldsymbol{p}(s)$, which includes $p(a \mid s)$ for each action $a$. In a board game, $s$ represents the board state, and $a$ denotes a move.

The input to the DNN is an $H \times W \times (2T + 1)$ image stack $\mathbf{x} \in \mathbb{R}^{H \times W \times (2T+1)}$, consisting of $2T + 1$ binary feature planes of size $H \times W$. Here, $H$ and $W$ are the dimensions of the board, and $T$ is the number of history planes. The first $T$ feature planes represent the occupancies of the first player's discs, where a feature value of 1 means that a disc occupies the corresponding cell, and 0 means it does not. The following $T$ feature planes represent the occupancies of the second player's discs. The final feature plane represents the color of the current player's disc, where 1 denotes the first player and $-1$ denotes the second player.

The convolutional layer divides the image stack $\mathbf{x}$ into $P \times P$ patches with stride str and padding pad, and generates the patch embeddings $\mathbf{z}_{\text{patch}}$. The patch size $P$ corresponds to the kernel size of the convolutional layer. The sequence of patch embeddings $\mathbf{z}_{\text{patch}}$ is defined in Eq. (1) as follows:

$$\mathbf{z}_{\text{patch}} = [\mathbf{x}_p^0 \mathbf{E}; \dots; \mathbf{x}_p^i \mathbf{E}; \dots; \mathbf{x}_p^{N_p-1} \mathbf{E}], \tag{1}$$

where $N_p$ is the number of patches, $\mathbf{x}_p^i \in \mathbb{R}^{P^2(2T+1)}$ is the $i$th flattened 2D patch, and $\mathbf{E}$ is a trainable embedding tensor with the shape $(P^2(2T + 1), D)$. Here, $N_p = (\lfloor H + 2\,\text{pad} - P\text{str} \rfloor + 1) \times (\lfloor W + 2\,\text{pad} - P\text{str} \rfloor + 1)$, where $H$ and $W$ are the board height and width, $P$ is the patch size, str is the stride, and pad is the padding used in the convolutional layer. The kernel of the convolutional layer acts as the tensor $\mathbf{E}$, which maps each patch to a $D$-dimensional embedding space. Here, $D$ is the embedding size.

To retain positional information, learnable 2D positional embeddings $\mathbf{E}_{\text{pos}}$ are added to the patch embeddings. These positional embeddings are scaled according to the board size and hyperparameters to match the size of the patch embeddings $\mathbf{z}_{\text{patch}}$. The resulting position-aware patch embeddings are given by Eq. (2):

$$\mathbf{z}_{\text{patch}}^{\text{pos}} = \mathbf{z}_{\text{patch}} + \mathbf{E}_{\text{pos}}. \tag{2}$$

The output size of the transformer encoder is determined by the number of input tokens. For Gomoku, where the action space is $HW$, AlphaViT requires $HW$ embeddings (*i.e.*, $N_p = HW$). To achieve this, the patch size $P$ is set to $2k + 1$, where $k$ is a non-negative

integer, stride str is set to 1, and padding pad is set to $\lfloor P/2 \rfloor$. Note that when $k > 0$, patches overlap, but the total number of patches remains $HW$. In contrast, for Othello, the action space is $HW + 1$ to include the pass move. Using the same parameters as Gomoku ($P = 2k + 1$, str $= 1$, and pad $= \lfloor P/2 \rfloor$), AlphaViT requires one additional token for the pass move. To address this, a learnable pass token $\mathbf{x}_{\text{pass}}$ is introduced by appending it after all patch embeddings (see Eq. (3)). To estimate the board value, a learnable value token $\mathbf{x}_{\text{value}}$ is prepended. Additionally, to enable AlphaViT to recognize different game types, a non-trainable one-hot game token $\mathbf{x}_{\text{game}}$, represented using one-hot encoding, is incorporated. These tokens are appended to the embeddings $\mathbf{z}_{\text{patch}}^{\text{pos}}$. As a result, the input embeddings $\mathbf{z}_0$ to the transformer encoder are defined in Eq. (3) as follows:

$$\mathbf{z}_0 = [\mathbf{x}_{\text{value}}; \mathbf{x}_{\text{game}}; \mathbf{z}_{\text{patch}}^{\text{pos}}; \mathbf{x}_{\text{pass}}]. \tag{3}$$

Positional embeddings are added to the patch embeddings before appending the pass, value, and game embeddings. This approach ensures that the positional embeddings can be scaled independently, without being affected by embeddings that do not inherently contain positional information.

The sequence $\mathbf{z}_0$ is fed into the transformer encoder, which consists of $L$ transformer encoder layers. The output of the final encoder layer $\mathbf{z}_L$ has shape $(HW + 3) \times D$. The first vector $\mathbf{z}_L^0$ derived from the value token is processed by the value head implemented as an MLP denoted as $\text{MLP}_v$. This head estimates the value $v$ using Eq. (4):

$$v = \tanh(\text{MLP}_v(\text{LN}(\mathbf{z}_L^0))), \tag{4}$$

where LN represents layer normalization. The tanh activation constrains the value within the range $(-1, 1)$, representing the state value, where 1 indicates a certain win and $-1$ a certain loss.

The vectors $\mathbf{z}_p = [\mathbf{z}_L^2, \dots, \mathbf{z}_L^{HW+2}]$ (index 0: value, 1: game, $2 \dots HW + 1$: patches, $HW + 2$: pass), derived from the board patches and the pass token, are processed by the policy head, which is implemented as another MLP ($\text{MLP}_p$). The $H \times W$ board positions and the pass move are flattened into one-dimensional indices $i \in \{0, 1, \dots, HW\}$. For $i < HW$, the move corresponds to board coordinates $(m, n)$ where $m = i \mod W$ is the column index, and $n = \lfloor i/W \rfloor$ is the row index. The policy head applies a shared MLP to each token, and a softmax over the resulting logits yields a probability vector $\boldsymbol{p}(s) \in \mathbb{R}^{HW+1}$ whose $i$th entry is $p(a_i \mid s)$. The special index $i = HW$ represents the pass action. Thus, the policy head output is defined in Eq. (5):

$$p(a_i \mid s) = \text{Softmax}(\text{MLP}_p(\text{LN}(\mathbf{z}_p)))_i, \tag{5}$$

where $a_i$ is the $i$th action, and the MLP is applied to each of the $HW + 1$ tokens. For Othello, the probability for the pass move is $p(a_{HW} \mid s)$. In Gomoku, this entry is masked out since the pass move is not allowed. In Connect 4, only the probabilities $\{p(a_i \mid s) \mid 0 \le i < W\}$ (i.e., moves ($m = i, n = 0$)) corresponding to valid column choices $i$ are used and renormalized. Note that $i$ is the column index in Connect 4. The agent drops the disc into column $i$, and the disc then falls to the lowest available row.

To decide on a move, AlphaViT employs MCTS with Upper Confidence Bound applied to Trees (UCT), using the value and move probabilities computed by the DNN. The MCTS algorithm used in AlphaViT is identical to that of AlphaZero, as detailed in 'AlphaZero'. The hyperparameters for AlphaViT are listed in 'Parameters'.

**AlphaViD and AlphaVDA** AlphaViT's DNN has a significant drawback: the size of the move probability vector (policy) is fixed by the transformer encoder's input size. To address this issue, AlphaViD and AlphaVDA incorporate both a transformer encoder and a decoder for calculating value and move probabilities, as shown in Fig. 3. In AlphaViD, the decoder receives input derived from the output of the encoder. In contrast, AlphaVDA employs learnable action embeddings as inputs to the decoder. The value is computed from the encoder's final output, while the move probabilities are computed from the final output of the decoder.

In AlphaViD and AlphaVDA, the input board is linearly embedded using a convolutional layer and fed into a transformer encoder, similar to AlphaViT. However, unlike AlphaViT, no pass token is included in the encoder input. The input embedding sequence is defined in Eq. (6) as follows:

$$\mathbf{z}_0 = [\mathbf{x}_{\text{value}}; \mathbf{x}_{\text{game}}; \mathbf{z}_{\text{patch}}^{\text{pos}}], \tag{6}$$

where the sequence consists of a value token, a game token, and patch embeddings. The estimated value is obtained from the value head that processes the output embedding corresponding to the value token from the last layer of the transformer encoder. In both AlphaViD and AlphaVDA, the encoder output is only used directly for value estimation, indicating that the number of patch embeddings $N_p$ need not match the action-space size.

The architecture of AlphaViD's DNN is shown at the left of Fig. 3. The DNN estimates the move probability vector using the transformer decoder and $\text{MLP}_p$. The input embeddings for the transformer decoder are derived from the outputs of the transformer encoder corresponding to the patch embeddings, which are further processed through a fully connected layer. The initial embeddings for the decoder input are defined in Eq. (7) as follows:

$$\mathbf{y}_0' = \text{MLP}([\mathbf{z}_L^2; \ldots; \mathbf{z}_L^{N_p+1}]), \quad \mathbf{y}_0' \in \mathbb{R}^{N_p \times D_d}, \tag{7}$$

where $D_d$ is the embedding size of the decoder. Since the embedding sequence size must match the input size of the transformer decoder, $\mathbf{y}_0'$ is interpolated to $\mathbf{y}_0 \in \mathbb{R}^{N_a \times D_d}$, where $N_a$ is the action space size. In this study, this step uses bilinear interpolation in the function *torch.nn.functional.interpolate*. This interpolation provides flexibility to adjust the action space size depending on the game type and board size. If an additional move such as a "pass" (*e.g.*, in Othello) is required, $N_a$ is set to $HW + 1$ and $\mathbf{y}_0'$ is resized accordingly *via* this interpolation. The additional embedding at the last position is assigned to the additional move (*e.g.*, the pass). Therefore, a dedicated pass token is not required for the encoder's input. Similar to the original transformer, the transformer decoder receives $\mathbf{y}_0$ as the target sequence and $\mathbf{z}_L$ as the memory (encoder output). Finally, $\text{MLP}_p$ calculates the

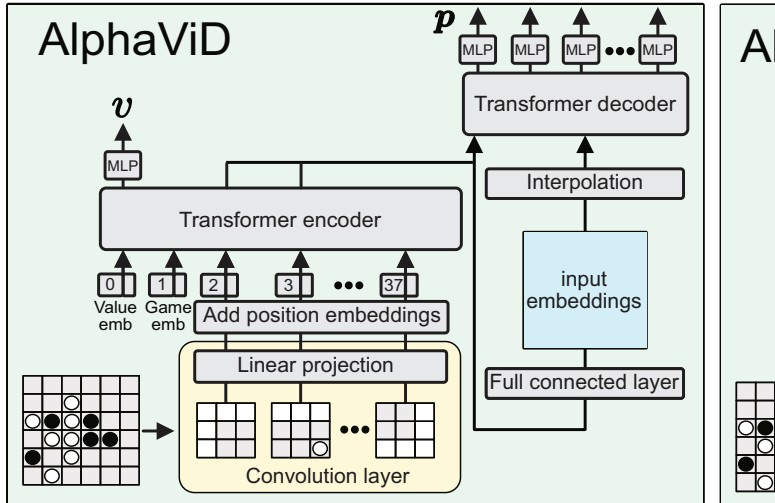
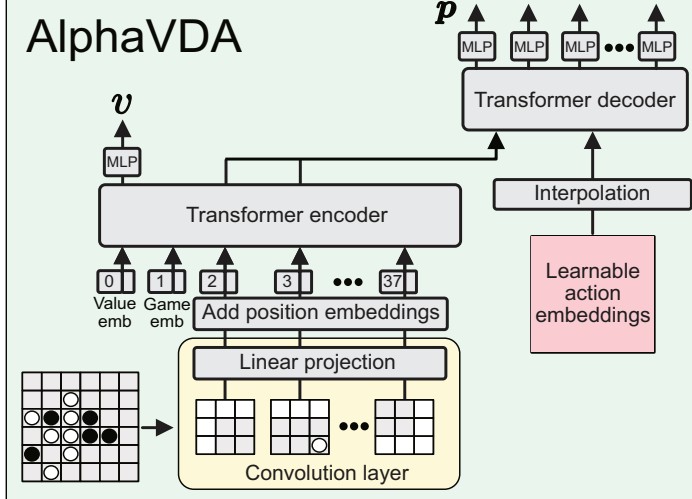

**Figure 3 Architecture diagrams of AlphaViD (left) and AlphaVDA (right).** The input board is divided into patches, embedded, and combined with value and game tokens (embeddings), then processed by a transformer encoder. For policy prediction, encoder outputs are sent to a transformer decoder *via* interpolation. In AlphaViD, decoder inputs are generated from board-derived embeddings *via* a fully connected layer; in AlphaVDA, learnable action embeddings are used. MLP heads output the final value and move probabilities.

move probabilities from the output of the last layer of the transformer decoder $\mathbf{y}_L$. The probability of action $a_i$ is given by Eq. (8):

$$p(a_i \mid s) = \text{Softmax}(\text{MLP}_p(\text{LN}(\mathbf{y}_L)))_i. \tag{8}$$

The architecture of AlphaVDA's DNN is shown at the right of Fig. 3. The DNN is similar to that of AlphaViD, but uses learnable embeddings $\mathbf{y}'_0$ as the initial embeddings for the transformer decoder. While the length of these learnable embeddings is fixed, they are interpolated to the decoder input $\mathbf{y}_0$ to match the action space size, ensuring compatibility and adaptability across different game configurations.

To decide on a move, AlphaViD and AlphaVDA employ MCTS with UCT, using the value and move probabilities predicted by their DNNs. The MCTS algorithm implemented in these agents is identical to that of AlphaZero, as detailed in 'AlphaZero'. The hyperparameters for AlphaViD and AlphaVDA are listed in 'Parameters'.

## EXPERIMENTAL SETUP

### Games

This study evaluates the proposed agents on six game variants (three different games, each played on two board sizes): Connect 4 (7 × 6; "Connect 4", and 5 × 4; "Connect 4 5 × 4"), Gomoku (9 × 9; "Gomoku", and 6 × 6; "Gomoku 6 × 6"), and Othello (8 × 8; "Othello", and 6 × 6; "Othello 6 × 6"). These games are two-player, deterministic, zero-sum games with perfect information. Connect 4 is a connection game played on a 7 × 6 board, published by Milton Bradley. The players take turns dropping discs onto the board. A player wins by forming a straight line of four discs horizontally, vertically, or diagonally. Connect 4 5 × 4 is a variant of Connect 4 on a 5 × 4 board. Gomoku is a connection game

in which players place stones on a board to form a straight line of five stones, either horizontally, vertically, or diagonally. This study uses a $9 \times 9$ board for Gomoku and a $6 \times 6$ board for Gomoku $6 \times 6$. Othello (also known as Reversi) is a two-player strategy game played on an $8 \times 8$ board. In Othello, the discs are white on one side and black on the other. Players take turns placing a disc with their assigned color facing up. During a game, discs of the opponent's color are flipped to the current player's color if they are in a straight line and bounded by the disc just placed and another disc of the current player's color. Othello $6 \times 6$ is played on a $6 \times 6$ board in this study. These games were selected (i) each has natural board-size variants ($5 \times 4 - 9 \times 9$), (ii) they are commonly used for benchmarking, and (iii) training can be completed on commodity GPUs. The description of the game rules is partly adapted from our previous publication (*Fujita, 2022*).

## Opponents

This study evaluates the performance of AlphaViT, AlphaViD, and AlphaVDA using five different AI methods: AlphaZero, two variants of MCTS labeled MCTS100 and MCTS400, Minimax, and Random. AlphaZero was trained using the method described in 'AlphaZero'. The MCTS methods (MCTS100 and MCTS400) were run with 100 and 400 simulations, respectively.

The details of MCTS are provided in the Supplemental Material (Sec. S1). In these MCTS methods, the child nodes are expanded on the fifth visit to a node. Minimax selects a move using the minimax algorithm based on the evaluation table described in the Supplemental Material (Sec. S2). The Random agent selects moves uniformly at random from the set of valid moves.

Previous studies indicate that vanilla MCTS with random rollouts scales poorly, remaining weaker than a shallow $\alpha-\beta$ search even with an increased number of simulations. For instance, in Connect-4, a UCT agent employing 10,000 random simulations achieved only a 19.8 % win rate against a depth-8 $\alpha-\beta$ opponent (*Scheiermann & Konen, 2023*). In contrast, incorporating domain-specific heuristics such as "decisive-move" pruning substantially improves MCTS performance, often yielding strength increases of one to two orders of magnitude (*Teytaud & Teytaud, 2010*; *Taylor & Stella, 2024*). The measurements in this study (the Supplemental Material, Sec. S3) corroborate these findings for MCTS, showing rapid Elo improvement up to approximately 400 simulations followed by diminishing returns. Therefore, this study adopts 100 and 400 simulations as the MCTS baselines, providing clearly separated strength levels while maintaining reasonable computational requirements.

## Software

AlphaViT, AlphaViD, AlphaVDA, the opponents, and the board games were implemented in Python, using NumPy for linear algebra operations and PyTorch for DNNs. The source code and the raw data are available on GitHub at https://github.com/KazuhisaFujita/AlphaViT for reproducibility and further extensions of this work.

### Hardware

All experiments were conducted on multiple custom-built PCs. The agents were run on a variety of consumer CPUs, including AMD Ryzen 9 9900X and Intel Core i7-14700K, Core i9-13900K, and Core i9-12900K processors. The GPUs used for the experiments were NVIDIA GeForce RTX 4060 Ti (16 GB), RTX 3060 (12 GB), RTX 2070 Super, and RTX 2070. Each machine was equipped with 64 GB of RAM and two GPUs, and the experiments were run on Debian Linux. Model training and evaluation were distributed across multiple GPUs on a single machine using data parallelism. No cloud-based or high-performance computing clusters were used, and all computations were performed on local custom-built hardware. For example, one of the machines used for training and evaluation was equipped with an Intel Core i9-13900K CPU, 64 GB of RAM, and two NVIDIA GeForce RTX 4060 Ti (16 GB) GPUs.

## RESULTS

The results section presents the performance and characteristics of AlphaViT, AlphaViD, and AlphaVDA, which were trained on different games (Connect 4, Gomoku, and Othello) with two board sizes (large and small). The main architectural difference between these agents lies in the number of encoder layers, which directly affects their learning capacity. Table 1 shows the number of parameters for each agent configuration. Each agent was tested with different numbers of encoder layers, denoted by 'L' followed by a number (*e.g.*, L1, L4, L5, L8). The number of parameters ranges from 11.2 to 19.9 million, increasing with encoder depth. For comparison, the AlphaZero agent, which serves as the baseline, has 7.1 million parameters.

The primary architectural variable is the number of transformer encoder layers. Throughout this article, four layers for AlphaViT and one layer for AlphaViD and AlphaVDA ($\approx$11 million parameters) are designated as the baseline configuration, which is also referred to as the shallower configuration in contrast to the deeper variants described later. By contrast, deeper encoders use exactly four additional layers (*e.g.*, L8 for AlphaViT, L5 for AlphaViD/AlphaVDA, $\approx$20 million parameters). Here, "deeper" is used only in this *relative* sense and does not imply a universal threshold on the number of layers.

Each AI agent was trained on specific games with different board sizes. Table 2 categorizes the agents based on the games on which they were trained and the board sizes used during training. The first group includes agents trained on a single game with a large board, denoted as LB. The second group consists of agents trained on a single game with a small board, denoted as SB. The third group comprises agents simultaneously trained on multiple games, including Connect 4, Gomoku, and Othello, with large boards, denoted as Multi. The agents in the third group were trained on the three games and can play these three games using a single DNN. In other words, they do not specialize in a specific game. This diversity of training settings allows us to evaluate the agents' adaptability and generalization capabilities across different game domains.

A brief sanity-check comparing our baseline AlphaZero to AlphaZeroGP, which is a simplified implementation using Wu's global-pooling method, is provided in the Supplemental Note: AlphaZeroGP.

**Table 1 Encoder layer variations and parameter sizes in AI agents.**

| AI agent | Number of encoder layers | Number of parameters |
|---|---|---|
| AlphaViT L4 | 4 | 11.2M |
| AlphaViD L1 | 1 | 11.5M |
| AlphaVDA L1 | 1 | 11.3M |
| AlphaViT L8 | 8 | 19.6M |
| AlphaViD L5 | 5 | 19.9M |
| AlphaVDA L5 | 5 | 19.8M |
| AlphaZero | – | 7.1M |

**Table 2 Board size and game variations in AI agent training.**

| AI agents | Game | Board size |
|---|---|---|
| AlphaViT LB, AlphaViD LB, AlphaVDA LB | One specific game | Large |
| AlphaViT SB, AlphaViD SB, AlphaVDA SB | One specific game | Small |
| AlphaViT Multi, AlphaViD Multi, AlphaVDA Multi | Connect 4, Gomoku, Othello | Large |

## Baseline Elo ratings

**Objective and Setup.** Tables 3, 4, and 5 present Elo ratings of various AI agents across different games and board sizes. Elo ratings provide a standard measure of relative performance in two-player games and enable systematic comparison across agents. This study evaluated multiple variants of the proposed models, including agents trained on a single game with either large or small boards, and multiple games with large boards. Comparisons were conducted with other AI agents, including AlphaZero, MCTS with different numbers of simulations, Minimax, and a Random agent. The proposed agents and AlphaZero underwent 1,000 training iterations; each iteration consisted of self-play, augmentation, and an update step detailed in 'Training procedure'. The Elo ratings of all agents were initialized to 1,500 and were calculated through 50 round-robin tournaments between the agents. Each Elo rating is accompanied by a 95% confidence interval (95% CI); the calculation details are provided in the Supplemental Material (Sec. S4).

Note: Throughout this article, the term *strong performance* refers to benchmark-relative performance. An agent is considered strong if its Elo rating exceeds the baseline AlphaZero's Elo rating minus 100 (as shown in Tables 3, 4, and 5). Thus, *strong performance* does not denote an absolute numerical threshold, but rather performance at the level of AlphaZero.

**Highlights across games.** Single-task AlphaViT L4 approaches AlphaZero's performance across all evaluated games, remaining within approximately 270 Elo points of AlphaZero. Single-task AlphaViD L1 and AlphaVDA L1 achieve AlphaZero-level performance only on smaller boards (Connect 4 $5 \times 4$ and Gomoku $6 \times 6$), yet they experience substantial drops in Elo ratings on larger boards. Multitask AlphaViT L4 (AlphaViT L4 Multi) demonstrates competitive performance on large boards and closely approaches AlphaZero (within 200 Elo points). Conversely, multitask AlphaViD L1 and AlphaVDA L1 variants

**Table 3 Elo ratings of AI agents for Connect 4 variants.**

| Agent | Connect 4 | | | Connect 4 5 × 4 | | |
|---|---|---|---|---|---|---|
| | Elo | Δ Elo | 95% CI | Elo | Δ Elo | 95% CI |
| AlphaZero | **2,114** | – | [2,073–2,159] | **1,769** | – | [1,747–1,797] |
| AlphaViT L4 LB | 1,846 | −267.4 | [1,799–1,902] | 1,517 | −252 | [1,480–1,554] |
| AlphaViD L1 LB | 1,739 | −374.7 | [1696–1783] | 1,462 | −306.9 | [1,426–1,495] |
| AlphaVDA L1 LB | 1,746 | −367.3 | [1,707–1,788] | 1,507 | −262.1 | [1,472–1,539] |
| AlphaViT L4 SB | 1,202 | −911.6 | [1,152–1,252] | **1,751** | −18.07 | [1,722–1,777] |
| AlphaViD L1 SB | 1,177 | −936.4 | [1,135–1,214] | **1,764** | −4.512 | [1,741–1,793] |
| AlphaVDA L1 SB | 1,251 | −862.5 | [1,201–1,301] | **1,789** | 19.98 | [1,763–1,818] |
| AlphaViT L4 Multi | 1,950 | −164.0 | [1,917–1,988] | 1,402 | −366.4 | [1,364–1,437] |
| AlphaViD L1 Multi | 1,745 | −369.2 | [1,706–1,785] | 1,317 | −451.4 | [1,271–1,361] |
| AlphaVDA L1 Multi | 1,669 | −444.7 | [1,627–1,710] | 1,305 | −463.3 | [1,266–1,343] |
| MCTS400 | 1,516 | −836.1 | [1,467–1,560] | 1,563 | −205.1 | [1,530–1,599] |
| MCTS100 | 1,278 | −598.2 | [1,230–1,316] | 1,502 | −266.9 | [1,468–1,533] |
| Minimax | 1,012 | −1,102 | [970.6–1,053] | 1,316 | −453 | [1,277–1,346] |
| Random | 755.4 | −1,358 | [722.5–788.9] | 1,038 | −730.4 | [992.4–1,082] |

Note:
Elo ratings of AI agents for Connect 4 variants. Bolded ratings indicate agents whose Elo rating is within 100 points of AlphaZero. Elo ratings were calculated from 50 round-robin tournaments beginning with an initial rating of 1,500, including differences relative to AlphaZero (Δ Elo) and corresponding 95% confidence intervals.

**Table 4 Elo ratings of AI agents for Gomoku variants.**

| Agent | Gomoku | | | Gomoku 6 × 6 | | |
|---|---|---|---|---|---|---|
| | Elo | Δ Elo | 95% CI | Elo | Δ Elo | 95% CI |
| AlphaZero | **2,038** | – | [1,982–2,091] | **1,807** | – | [1,788–1,830] |
| AlphaViT L4 LB | **1,966** | −72.4 | [1,924–2,012] | 1,645 | −162 | [1,611–1,678] |
| AlphaViD L1 LB | 1,698 | −340 | [1,659–1,744] | 1,509 | −298.3 | [1,476–1,546] |
| AlphaVDA L1 LB | 1,567 | −471.5 | [1,521–1,614] | 1,261 | −546.3 | [1,219–1,297] |
| AlphaViT L4 SB | 1,553 | −485.7 | [1,500–1,605] | **1,764** | −42.78 | [1,745–1,786] |
| AlphaViD L1 SB | 1,519 | −519.6 | [1,466–1,565] | **1,779** | −27.43 | [1,760–1,798] |
| AlphaVDA L1 SB | 977.2 | −1,061 | [943.8–1,011] | **1,771** | −35.96 | [1,751–1,792] |
| AlphaViT L4 Multi | **2,024** | −13.93 | [1,985–2,069] | 1,656 | −150.8 | [1,626–1,688] |
| AlphaViD L1 Multi | 1,530 | −508.7 | [1,484–1,576] | 1,378 | −428.8 | [1,339–1,416] |
| AlphaVDA L1 Multi | 1,570 | −467.8 | [1,517–1,622] | 1,170 | −636.9 | [1,130–1,209] |
| MCTS400 | 1,161 | −877.4 | [1,116–1,201] | 1,680 | −126.3 | [1,651–1,709] |
| MCTS100 | 1,229 | −809.5 | [1,180–1,274] | 1,380 | −426.3 | [1,343–1,418] |
| Minimax | 1,472 | −566.4 | [1,425–1,518] | 1,348 | −458.6 | [1,309–1,384] |
| Random | 696.8 | −1,341 | [673.9–719.2] | 851.9 | −954.9 | [820.8–884.9] |

Note:
Elo ratings of AI agents for Gomoku variants. Bolded ratings indicate agents whose Elo rating is within 100 points of AlphaZero. Elo ratings were calculated from 50 round-robin tournaments beginning with an initial rating of 1,500, including differences relative to AlphaZero (Δ Elo) and corresponding 95% confidence intervals.

**Table 5 Elo ratings of AI agents for Othello variants.**

| Agent | Othello | | | Othello 6x6 | | |
|---|---|---|---|---|---|---|
| | Elo | Δ Elo | 95% CI | Elo | Δ Elo | 95% CI |
| AlphaZero | **1,996** | – | [1,953–2,048] | **2,034** | – | [1,986–2,081] |
| AlphaViT L4 LB | **2,017** | 20.55 | [1,973–2,065] | 1,819 | −215.5 | [1,776–1,863] |
| AlphaViD L1 LB | **1,896** | −99.95 | [1,854–1,941] | 1,585 | −449.5 | [1,539–1,631] |
| AlphaVDA L1 LB | 1,669 | −327.2 | [1,620–1,718] | 1,368 | −666.4 | [1,322–1,415] |
| AlphaViT L4 SB | 1,482 | −513.9 | [1,435–1,528] | **1,963** | −71.16 | [1,919–2,013] |
| AlphaViD L1 SB | 1,184 | −812.4 | [1,132–1,228] | 1,803 | −231.6 | [1,757–1,854] |
| AlphaVDA L1 SB | 1,111 | −885.3 | [1,063–1,159] | 1,855 | −179.6 | [1,799–1,911] |
| AlphaViT L4 Multi | **1,910** | −86.4 | [1,865–1,958] | 1,090 | −944.2 | [1,046–1,134] |
| AlphaViD L1 Multi | 1,668 | −328.7 | [1,626–1,709] | 1,308 | −726.4 | [1,264–1,354] |
| AlphaVDA L1 Multi | 1,578 | −418 | [1,529–1,626] | 991.1 | −1,043 | [938.4–1,034] |
| MCTS400 | 1,373 | −623.6 | [1,329–1,412] | 1,579 | −455 | [1,535–1,623] |
| MCTS100 | 1,194 | −802.7 | [1,148–1,241] | 1,367 | −667.6 | [1,324–1,412] |
| Minimax | 1,140 | −855.9 | [1,089–1,190] | 1,363 | −670.9 | [1,311–1,411] |
| Random | 782.8 | −1,213 | [744.8–816.5] | 874.2 | −1160 | [824–919] |

**Note:**
Elo ratings of AI agents for Othello variants. Bolded ratings indicate agents whose Elo rating is within 100 points of AlphaZero. Elo ratings were calculated from 50 round-robin tournaments beginning with an initial rating of 1,500, including differences relative to AlphaZero (Δ Elo) and corresponding 95% confidence intervals.

exhibit significantly lower performance. Note that the multitask models were not trained on small board configurations and therefore underperform on these board sizes.

**Connect 4 (Table 3).** For standard Connect 4, AlphaViT L4 Multi most closely approaches AlphaZero (Δ Elo ≈ −160). AlphaViT L4 LB is moderately weaker (Δ Elo ≈ −260), whereas AlphaViD L1 LB and AlphaVDA L1 LB lag behind significantly (Δ Elo < −300). On the smaller 5 × 4 board, all single-task variants (AlphaViT L4 SB, AlphaViD L1 SB, AlphaVDA L1 SB) are competitive with AlphaZero.

**Gomoku (Table 4).** For the large board, only AlphaViTs (L4 LB and Multi) attain performance comparable to AlphaZero, with overlapping confidence intervals. AlphaViD and AlphaVDA variants (L1 LB and Multi) substantially lag behind (Δ Elo < −300). On the smaller board, SB variants from all architectures effectively approach AlphaZero-level performance.

**Othello (Table 5).** On the large board, AlphaViT L4 LB numerically surpasses AlphaZero (Δ Elo ≈ +20), though overlapping confidence intervals prevent definitive conclusions. AlphaViD L1 LB and AlphaViT L4 Multi achieve performance near that of AlphaZero, with Δ Elo ≈ −100. AlphaVDA L1 LB lags behind significantly (Δ Elo < −300). On the smaller board, AlphaViT L4 SB closely approaches AlphaZero (Δ Elo ≈ −70). AlphaViD L1 SB lags behind (Δ Elo ≈ −230), whereas AlphaVDA L1 SB also falls short (Δ Elo ≈ −180), remaining weaker than AlphaZero. Interestingly, AlphaViT L4 LB, despite being trained only on the large board, achieves performance comparable to AlphaViD L1 SB and

AlphaVDA L1 SB on the smaller board. This suggests that knowledge may transfer from larger to smaller boards.

## Conclusion

In summary, AlphaViT L4 and AlphaViT L4 Multi consistently achieve or closely approach AlphaZero-level strength across games, even without game-specific fine-tuning. In contrast, AlphaViD L1 and AlphaVDA L1 achieve AlphaZero-level performance only on smaller boards. The strong results for multitask agents (AlphaViT L4 Multi) suggest that multitask training will not significantly hinder performance. The Elo ratings presented in Table 3, 4, and 5 serve as a baseline for the subsequent experiments described in the following sections. Further analysis of AlphaZero's architectural variants, including deeper networks, fine-tuning, and optimizer choices, is presented in the Supplemental Material (Sec. S5).

## Variation of Elo rating over training iterations

**Setup.** Figure 4 illustrates the progression of Elo ratings for AlphaViT, AlphaViD, AlphaVDA, and AlphaZero across multiple training iterations for large and small board configurations in three games: Connect 4, Gomoku, and Othello. Elo ratings were recorded at every iteration from 1 to 10, subsequently every 20 iterations up to iteration 100, and thereafter at intervals of 100 iterations until iteration 3,000 for large board configurations and 1,000 for small board configurations. The different evaluation intervals were used because Elo ratings change more rapidly in the early training phase. The shaded areas around each Elo curve indicate the 95% confidence intervals computed through bootstrapping, as detailed in the Supplemental Material (Sec. S4). One iteration corresponds to a complete cycle of self-play data generation, data augmentation, and a single update of the neural network (see 'Training procedure'). The opponents used for evaluation were identical to those described in previous experiments: AlphaViT L4 LB, AlphaViD L1 LB, AlphaVDA L1 LB, AlphaViT L4 SB, AlphaViD L1 SB, AlphaVDA L1 SB, AlphaViT L4 Multi, AlphaViD L1 Multi, AlphaVDA L1 Multi, AlphaZero, MCTS400, MCTS100, Minimax, and Random agents. The Elo ratings of these opponents were fixed as listed in Tables 3, 4, and 5. Evaluated agents' Elo ratings were initialized to 1,500 and then calculated through 40-game matches against each opponent, consisting of 20 games as the first player and 20 games as the second player.

**Large board configurations.** Across all tested large board games, AlphaViT, AlphaViD, and AlphaVDA variants demonstrate rapid Elo improvement during the initial training phase, achieving approximately 80–90% of their maximum Elo within the first 300–500 iterations. In Gomoku, the single-task agents peak near 1,000 iterations and then exhibit a moderate decline. In Othello, the growth rate slows markedly after roughly 300 iterations, although the Elo ratings continue to increase slightly thereafter. Multitask-trained agents (AlphaViT Multi, AlphaViD Multi, AlphaVDA Multi) show learning speeds comparable to those of single-task models, confirming that multitask training does not adversely affect initial convergence. Notably, single-game-trained Gomoku models (AlphaViT L4 LB,

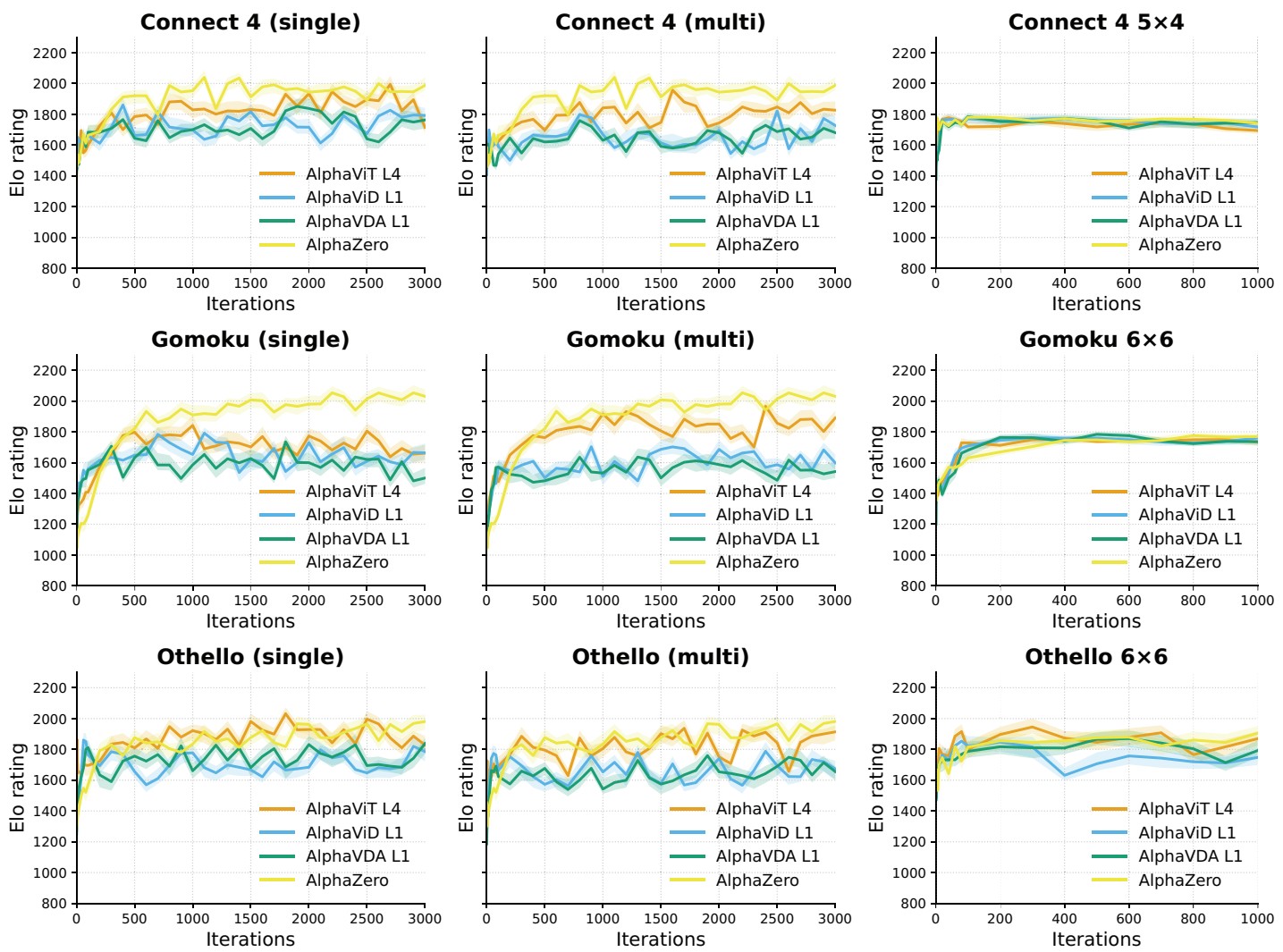

**Figure 4 Elo rating progression over training iterations for AlphaViT, AlphaViD, and AlphaVDA in large board configurations (Connect 4, Gomoku, and Othello) and small board configurations (Connect 4 5 × 4, Gomoku 6 × 6, and Othello 6 × 6).** The left and middle columns show single- and multi-game-trained agents for large board configurations, respectively, while the right column shows single-game-trained agents for the small board configurations. Solid lines represent Elo ratings calculated directly from aggregated game outcomes without employing bootstrapping, while shaded bands correspond to the 95% confidence intervals around these Elo ratings.

AlphaViD L1 LB, AlphaVDA L1 LB) exhibit moderate Elo declines after reaching peak performance, indicating potential overfitting in later training stages. Conversely, multitask-trained Gomoku models (AlphaViT L4 Multi, AlphaViD L1 Multi, AlphaVDA L1 Multi) remain stable without a pronounced decline, highlighting multitask learning as a potentially effective approach for mitigating overfitting.

**Small board configurations.** Agents trained on smaller board sizes exhibit notably faster convergence, attaining peak Elo ratings within 300 iterations (≈100 for Connect 4 5 × 4). After this rapid convergence, in Connect 4 5 × 4 and Gomoku 6 × 6, Elo ratings remain stable with minimal fluctuation, suggesting that small board training reliably and quickly

yields robust and stable agents. However, in Othello 6 × 6, Elo ratings exhibit slight fluctuations of approximately ±100.

**Insights and implications.** These findings clearly demonstrate that AlphaViT L4, AlphaViD L1, and AlphaVDA L1 quickly achieve high-performance levels on small boards. However, achieving similar performance stability on larger board configurations remains more challenging. The observed plateau in Elo improvement on large boards implies that simply extending training iterations beyond a certain point (around 1,000 iterations) yields limited additional benefit. Therefore, to achieve further performance improvements on larger boards, enhancing model complexity—such as employing deeper transformer encoders—rather than solely prolonging training duration may be necessary.

### Effect of transformer encoder depth on performance

In this subsection, the performance of the agents with deeper encoders, namely AlphaViT L8, AlphaViD L5, and AlphaVDA L5, is evaluated. Figure 5 shows the variations in Elo ratings for these agents over the iterations on large and small boards. All experiments followed the evaluation protocol described in the Setup paragraph of 'Variation of Elo rating over training iterations'.

For large board configurations, the Elo ratings of agents with deeper encoders gradually stabilize between roughly 1,000 and 2,000 iterations. This trend indicates a more protracted improvement phase compared to their shallower counterparts (AlphaViT L4, AlphaViD L1, and AlphaVDA L1). Conversely, in small board configurations, Elo ratings, as well as their 95% confidence intervals, converge more rapidly, similar to those of the shallow encoder agents. This suggests that in less complex game environments, the additional depth of encoders offers little substantial benefit.

Table 6 lists the mean Elo ratings calculated from iterations 2,100–3,000 for large board configurations. This highlights the performance gains achieved through the increased encoder depth, with AlphaViD and AlphaVDA demonstrating the most notable improvements. For example, AlphaViD L5 Multi and AlphaVDA L5 Multi achieve gains of +286 and +231 Elo points, respectively, in Gomoku, indicating significant performance enhancement.

The ratio of Elo ratings between the deeper and baseline DNNs, illustrated in Fig. 6, further substantiates these findings. Across all evaluated games, deeper architectures generally exhibit superior performance, with AlphaViD Multi and AlphaVDA Multi achieving the highest ratios. Specifically, in Gomoku, these agents achieve Elo ratios of 1.177 and 1.149, respectively. They also outperform the baseline models in Connect 4, with Elo ratios of 1.108 and 1.130, respectively. While single-game-trained AlphaViD and AlphaVDA show marked gains in Connect 4 and Othello, their performance improvements in Gomoku are modest. In contrast, both single- and multi-game-trained AlphaViT variants exhibit relatively smaller improvements than AlphaViD and AlphaVDA. In conclusion, these results collectively demonstrate that increasing the depth of the transformer encoder layers positively influences agent performance across various

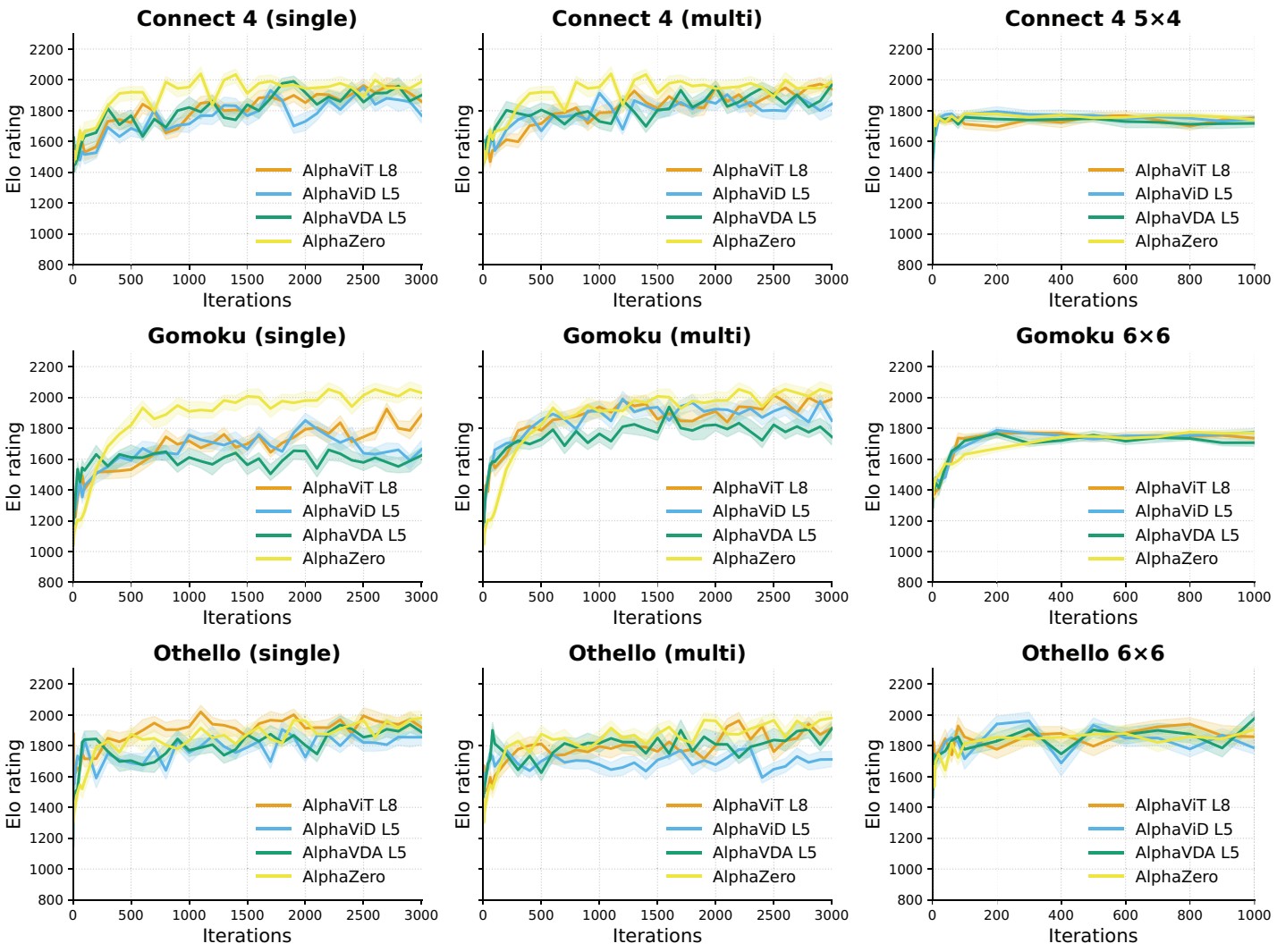

**Figure 5 Elo trajectories for the deep configurations. The left and the middle columns show single- and multi-game-trained agents for large board configurations, respectively, while the right column shows single-game-trained agents for small board configurations.** Solid lines represent Elo ratings calculated directly from aggregated game outcomes without employing bootstrapping, while shaded bands correspond to the 95% confidence intervals around these Elo ratings.

game types and board configurations. This effect is particularly pronounced for AlphaViD and AlphaVDA, especially in larger and more complex game settings.

## Effect of fine-tuning from small board games

Figure 7 illustrates the Elo ratings of AlphaViT L8, AlphaViD L5, and AlphaVDA L5 with fine-tuned and non-fine-tuned (randomly initialized) DNNs across three games: Connect 4, Gomoku, and Othello. The fine-tuned DNNs were initialized using weights from the single-task DNNs that had been trained for 200 iterations on the small board configuration. For example, the fine-tuned DNN for Connect 4 was initialized with weights from the DNN trained on Connect 4 5 × 4. In contrast, agents with non-fine-tuned DNNs were initialized with random weights. All experiments followed the evaluation protocol

**Table 6 Mean Elo Ratings from 2,100 to 3,000 iterations.**

| Game | Connect 4 (Δ Elo) | Gomoku (Δ Elo) | Othello (Δ Elo) |
|---|---|---|---|
| AlphaZero | **1,955** (−) | **2,019** (−) | **1,925** (−) |
| AlphaViT L4 LB | **1,870** (−) | 1,703 (−) | **1,890** (−) |
| AlphaViT L8 LB | **1,915** (+45) | 1,804 (+101) | **1,942** (+52) |
| AlphaViD L1 LB | 1,747 (−) | 1,630 (−) | 1,726 (−) |
| AlphaViD L5 LB | 1,851 (+104) | 1,682 (+52) | **1,843** (+117) |
| AlphaVDA L1 LB | 1,738 (−) | 1,569 (−) | 1,749 (−) |
| AlphaVDA L5 LB | **1,894** (+156) | 1,596 (+27) | **1,884** (+135) |
| AlphaViT L4 Multi | 1,827 (−) | 1,835 (−) | **1,848** (−) |
| AlphaViT L8 Multi | **1,909** (+82) | **1,947** (+112) | **1,883** (+35) |
| AlphaViD L1 Multi | 1,658 (−) | 1,616 (−) | 1,669 (−) |
| AlphaViD L5 Multi | 1,838 (+180) | 1,902 (+286) | 1,704 (+35) |
| AlphaVDA L1 Multi | 1,666 (−) | 1,556 (−) | 1,670 (−) |
| AlphaVDA L5 Multi | **1,884** (+218) | 1,787 (+231) | **1,833** (+163) |

Note:
Mean Elo ratings from 2,100 to 3,000 iterations for each agent. Δ Elo indicates the difference from a shallower model. Bolded ratings indicate agents whose Elo rating is within 100 points of AlphaZero.

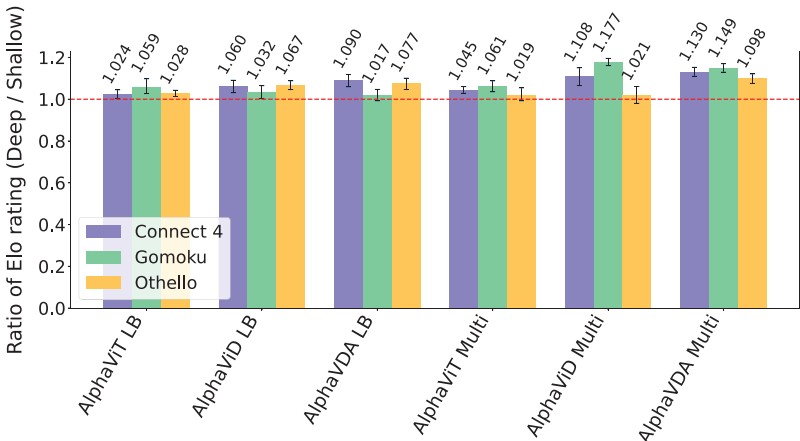

**Figure 6 The ratio of Elo ratings between deeper and baseline DNNs for AlphaViT, AlphaViD, and AlphaVDA across Connect 4, Gomoku, and Othello.** Ratios greater than 1.0 indicate superior performance of agents with deeper DNNs compared to those with baseline DNNs. The error bars represent 95% CIs calculated through bootstrapping (see the Supplemental Material, Sec. S4).

described in the Setup paragraph of 'Variation of Elo rating over training iterations'. For agents with non-fine-tuned (randomly initialized) DNNs, the Elo ratings previously reported in 'Effect of transformer encoder depth on performance' were reused.

The results demonstrate that agents with fine-tuned DNNs consistently achieve higher Elo ratings than those with randomly initialized DNNs for Gomoku and Othello. The performance difference is especially pronounced during the early iterations; in this phase, fine-tuned DNNs improve rapidly and then stabilize for Gomoku and Othello. For Gomoku, the advantage of fine-tuning is clear, as the agents with fine-tuned DNNs

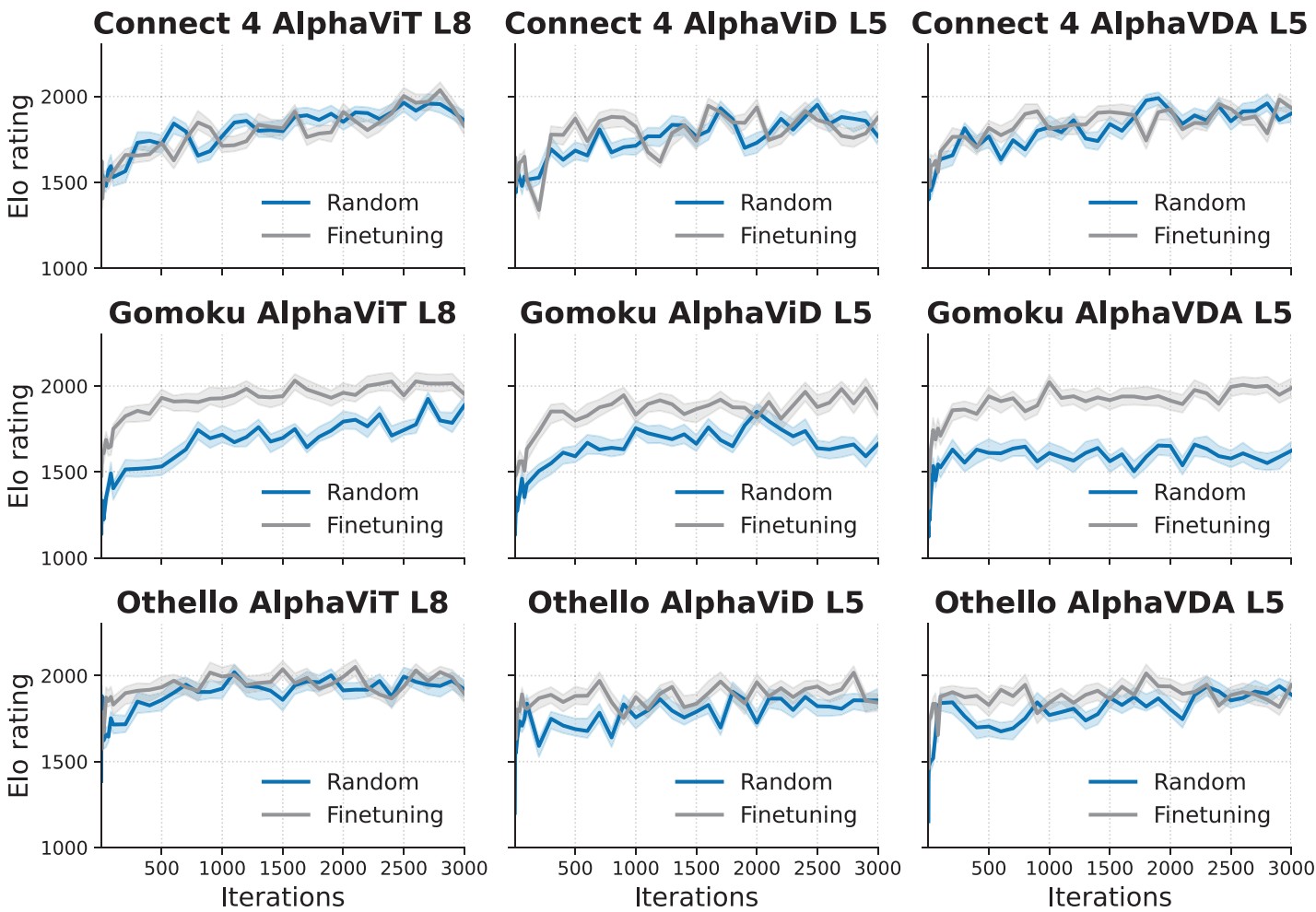

**Figure 7** **Elo rating progression over training iterations of the agents (AlphaViT L8, AlphaViD L5, and AlphaVDA L5) with fine-tuned and randomly initialized DNNs for three board games (Connect 4, Gomoku, and Othello).** The three columns show the results for AlphaViT L8, AlphaViD L5, and AlphaVDA L5, respectively. The three rows show the results for Connect 4, Gomoku, and Othello. The shaded bands represent the 95% confidence intervals around the Elo ratings, calculated through bootstrapping (see the Supplemental Material, Sec. S4).

consistently outperform those with non-fine-tuned DNNs throughout the training process. The plateau appears in AlphaViD L5 and AlphaVDA L5 for Gomoku between approximately 1,000 and 2,000 iterations. However, the performance improves after 2,000 iterations. For Connect 4, fine-tuning yields little or no improvement in Elo.

## DISCUSSION

The results of this study also relate to those of *Soemers et al. (2021, 2023)*, who highlighted the importance of transferring trained policies and value functions across games with varying board sizes and action spaces. Similarly, the proposed agents, equipped with DNNs that utilize weights either fine-tuned from a small board game or trained simultaneously on multiple games, demonstrate enhanced gameplay skills. This suggests that knowledge obtained from small board games and other games is used efficiently. The ability to train

on multiple games concurrently may also help these agents avoid overfitting, further enhancing their generalization capabilities.

As shown in Tables 3, 4, and 5, AlphaZero achieves the top performance with the smallest number of parameters in some cases, particularly for games with larger boards. This likely stems from the strong inductive bias inherent in its convolutional and residual network architecture, which is well-suited for single games and enables it to easily extract local and spatial features (He et al., 2016; Battaglia et al., 2018). Remarkably, this architecture also allows some flexibility in board size when game rules remain unchanged. For example, Wu's KataGo preserves the convolutional backbone while introducing global pooling layers—calculating channel-wise mean, scaled mean, and maximum—to both the trunk and heads, enabling a unified network to play Go from $9 \times 9$ through $19 \times 19$ without increasing the number of parameters (Wu, 2019).

However, the strong inductive biases of convolutional neural networks (CNNs) may limit adaptability when game rules or mechanics significantly change. Soemers et al. (2021) systematically demonstrated that fully convolutional networks can effectively transfer learning across different board sizes or minor variants of the same game but struggle significantly when transferring between games with substantially different rules or mechanics, resulting in poor or negative transfer performance even after fine-tuning. In contrast, transformer-based agents sacrifice some of this inductive bias in favor of greater flexibility (Dosovitskiy et al., 2021; d'Ascoli et al., 2022). This enables them to handle multiple games and variable board sizes at the cost of an increase in parameter count. In this study, when deep transformer encoders were employed and model parameters of AlphaViT, AlphaViD, and AlphaVDA were significantly larger than those of AlphaZero's DNN, their performance approached, and in Othello even surpassed, that of AlphaZero. This trade-off highlights a key design consideration: models with strong inductive bias can achieve higher efficiency in specialized domains, whereas more general architectures require additional capacity to compensate for their weaker assumptions (d'Ascoli et al., 2022).

Although the proposed agents with transformer-based DNNs are slightly weaker than AlphaZero, the author hypothesizes that the proposed agents have the potential for further improvement through enhanced training techniques. For example, a properly scheduled learning-rate decay during training iterations (Silver et al., 2016, 2017, 2018) and warm-start methods (Wang, Preuss & Plaat, 2020, 2021) are expected to improve performance. Increasing the number of self-plays per iteration may also be effective.

All experiments were carried out on affordable, consumer-grade GPUs (RTX 4060 Ti 16 GB and RTX 3060 12 GB) rather than on datacenter accelerators. Due to the significant memory requirements for training transformer architectures, even moderately sized boards, such as Gomoku $9 \times 9$ and Othello, approached the memory limits of these GPUs. To address this issue, automatic mixed-precision training with *torch.cuda.amp.autocast()* was enabled, so that most tensor operations run in FP16/BF16, while numerically sensitive layers and master weights remain in FP32. Additionally, data-parallel training was enabled across multiple GPUs within a single custom-built PC to further reduce per-device memory usage. These practical choices emphasize the reproducibility of AlphaViT (see the

Supplemental Material, Sec. S6) but also restrict experimentation with deeper transformer encoders and larger boards, such as 19 × 19 Go. Nevertheless, the proposed architecture inherently supports variable input sizes, and the fine-tuning experiments (effect of fine-tuning from small board games) indicate that weights pre-trained on small board games effectively generalize to larger board configurations. Consequently, when more powerful hardware is available, it will be feasible to first train agents with larger transformer encoders on small boards and then efficiently fine-tune, rather than training from scratch, to extend the proposed agents to full-sized boards and more complex games.

In future work, the author plans to extend these architectures to a broader range of games, including those with more complex rules and stochastic elements. In addition, the author aims to incorporate the flexibility of ViT-based DNNs into other deep reinforcement learning frameworks, such as deep Q-networks, to create AI agents capable of playing a wider variety of games, including video games, with enhanced adaptability.

## CONCLUSION

This article introduces AlphaViT, AlphaViD, and AlphaVDA, novel game-playing AI agents that use ViT to overcome the limitations of AlphaZero. Unlike AlphaZero, which is restricted to fixed board sizes, the proposed agents demonstrate adaptability, handle different board sizes effectively, and exhibit flexibility across games. Furthermore, these agents can simultaneously train on and play multiple games, such as Connect 4, Gomoku, and Othello, within a single shared neural network. The performance of these multitask agents surpasses traditional game AI algorithms and, in some cases, approaches that of AlphaZero.

The results of this study demonstrate that AlphaViT, AlphaViD, and AlphaVDA outperform traditional methods such as Minimax and MCTS across all tested scenarios. Although AlphaZero remains the top performer in some cases, particularly for games with larger boards, the proposed agents exhibit competitive performance. AlphaViT L8 matches AlphaZero in Connect 4 and Othello. In Othello, the deeper versions of AlphaViD and AlphaVDA (L5) narrow the performance gap with AlphaZero but do not yet surpass it. Multigame-trained variants perform on par with or better than single-game-trained variants with deeper DNNs in Connect 4 and Gomoku, while remaining slightly behind in Othello.

AlphaViT, AlphaViD, and AlphaVDA show strong adaptability across different games and board sizes. The agents with DNNs trained on a single game often achieve performance comparable to traditional game algorithms, such as Minimax and MCTS, even when playing on board sizes on which they are not trained. Moreover, multi-game-trained agents frequently perform on par with or surpass their single-game-trained counterparts. The agents with fine-tuned DNNs trained on small board games achieve better performance than that of agents with non-fine-tuned DNNs in Gomoku and Othello, but show little or no improvement in Connect 4. In the case of Gomoku, pre-trained weights from small board games significantly accelerate convergence and enhance the final performance. This suggests effective knowledge transfer between

different board sizes, mirroring human learning processes, in which skills acquired in simpler variants (*e.g.*, $9 \times 9$ Go) can be applied to more complex versions (*e.g.*, $19 \times 19$ Go). Such adaptability suggests that the proposed agents may have significant potential for advancing the development of multitask AI.

Comparing the three proposed agents reveals that AlphaViT L4 outperforms both AlphaViD L1 and AlphaVDA L1, despite having a similar number of parameters. This difference in performance may be attributed to the smaller number of encoder layers used in AlphaViD and AlphaVDA. This observation is further supported by the fact that AlphaViD L5 and AlphaVDA L5 exhibit performance comparable to AlphaViT L4. The simpler architecture of AlphaViT, consisting solely of encoder layers, may lead to more efficient performance in certain games, even when it has fewer parameters than its variants. However, this simplicity constrains AlphaViT's flexibility, as its output size is fixed to the number of input embeddings, limiting its applicability to games beyond classic board games. In contrast, including a decoder layer in AlphaViD and AlphaVDA allows for dynamic adjustment of the policy vector size, providing greater adaptability to games with varying action spaces. This architectural flexibility makes AlphaViD and AlphaVDA versatile candidates for handling more complex games or environments with continuous action spaces.

## APPENDIX

### AlphaZero

AlphaZero integrates a deep neural network (DNN) with Monte Carlo Tree Search (MCTS), as illustrated in Fig. 1. The DNN processes input representing the current board state and the current player, producing an estimated state value and a move-probability vector. MCTS then uses these outputs to select the optimal move. This same framework is adopted by AlphaViT, AlphaViD, and AlphaVDA.

### Deep neural network in AlphaZero

AlphaZero's DNN predicts a value $v(s)$ and a move-probability vector $\boldsymbol{p}(s)$ with components $p(a \mid s)$ for each action $a$, given a state $s$. In the board game context, $s$ and $a$ represent the board state and the move, respectively. The DNN receives input representing the current board state and the current player's disc color. Figure A1 illustrates the DNN architecture, which consists of a *Body* (residual blocks) and two *Heads* (value and policy heads). The value head outputs the estimated state value $v(s)$, while the policy head produces the move probabilities $\boldsymbol{p}(s)$.

The input to the DNN is an $H \times W \times (2T + 1)$ image stack that contains $2T + 1$ binary feature planes of size $H \times W$. Here, $H \times W$ refers to the board size, and $T$ is the number of histories (previous board states). The first $T$ feature planes represent the occupancy of the player's discs, with a feature value of 1 indicating that a disc occupies the corresponding cell, and 0 otherwise. Similarly, the following $T$ feature planes represent the occupancy of the opponent's discs. The final plane encodes the current player, being filled with $+1$ when it is the first player's turn and with $-1$ otherwise.

## Monte Carlo tree search in AlphaZero

This subsection provides an explanation of the Monte Carlo Tree Search (MCTS) algorithm used in AlphaZero. Each node in the game tree represents a game state, and each edge $(s, a)$ represents a valid action from that state. The edges store a set of statistics: $\{N(s, a), W(s, a), Q(s, a), p(s, a)\}$, where $N(s, a)$ is the visit count, $W(s, a)$ is the cumulative value, $Q(s, a) = W(s, a)/N(s, a)$ is the mean value, and $p(s, a)$ is the move probability.

The MCTS for AlphaZero consists of four steps: *Select*, *Expand and Evaluate*, *Backup*, and *Play*. A simulation is defined as a sequence of *Select*, *Expand and Evaluate*, and *Backup* steps, repeated $N_{\text{sim}}$ times. *Play* is executed after $N_{\text{sim}}$ simulations.

In *Select*, the tree is searched from the root node $s_{\text{root}}$ to the leaf node $s_L$ at time step $L$ using a variant of the Polynomial Upper Confidence Trees (PUCT) algorithm. At each time step $t < L$, the selected action $a_t$ has the maximum score, as described in Eq. (9):

$$a_t = argmax_a(Q(s_t, a) + C_{\text{puct}}p(s_t, a)\frac{\sqrt{N(s_t)}}{1 + N(s_t, a)}), \tag{9}$$

where $N(s_t)$ is the number of parent visits and $C_{\text{puct}}$ is the exploration rate. In this study, $C_{\text{puct}}$ is constant, whereas in the original AlphaZero, $C_{\text{puct}}$ increases slowly with search time (*Silver et al., 2018*). Additionally, when the parent node is the root node, the node selection is performed using an $\varepsilon$-greedy algorithm based on the UCT scores, where $\varepsilon$ denotes the exploration probability.

In *Expand and Evaluate*, the DNN evaluates the leaf node and outputs $v(s_L)$ and $\boldsymbol{p}(s_L)$. If the leaf node is a terminal node, $v(s_L)$ is the color of the winning player's disc. The leaf node is expanded and each edge $(s_L, a)$ is initialized to $\{N(s_L, a) = 0, W(s_L, a) = 0, Q(s_L, a) = 0, p(s_L, a) = p(a \mid s_L)\}$.

In *Backup*, the visit counts and values are updated for each step $t \leq L$ during the backward pass. The visit count is incremented by 1, $N(s_t, a_t) \leftarrow N(s_t, a_t) + 1$, and the cumulative and average values are updated, $W(s_t, a_t) \leftarrow W(s_t, a_t) + v$, $Q(s_t, a_t) \leftarrow W(s_t, a_t)/N(s_t, a_t)$.

Finally, in *Play*, AlphaZero selects the action corresponding to the most visited edge from the root node.

## TRAINING PROCEDURE

AlphaViT, AlphaViD, AlphaVDA, and AlphaZero share the same three-stage training loop: *Self-play*, *Augmentation*, and *Update*. One complete cycle of the three stages is called an **iteration** and is repeated $N_{\text{iter}}$ times. This training algorithm is a modified version of the original AlphaZero, adapted for a single-machine setting.

During the *Self-play* phase, an agent plays against itself $N_{\text{self}}$ times. For the first $T_{\text{opening}}$ turns, actions are stochastically selected among the valid moves according to the softmax policy defined in Eq. (10):

$$p(a \mid s) = \exp(N(s, a)/\tau)/\sum_b \exp(N(s, b)/\tau), \tag{10}$$

where $\tau$ is a temperature parameter that controls the exploration. This stochastic exploration enables the agent to explore new and potentially better actions. After $T_{\text{opening}}$, the most visited action is selected. During *Self-play*, the board states, game outcomes (winners), and the search probabilities are recorded. The search probabilities represent the probabilities of selecting valid moves at the root node in MCTS.

In the *Augmentation* phase, the dataset derived from *Self-play* is augmented by introducing symmetries specific to the game variant (*e.g.*, two symmetries for Connect 4 and eight for Othello and Gomoku). This augmented data is added to a queue with a capacity of $N_{\text{queue}}$ states to form the training dataset.

For the first update iteration, the training data queue is filled with data generated by self-play using MCTS100, which is then augmented. In subsequent iterations, new data generated by *Self-play* are added to the training data queue. To simultaneously learn multiple games, a separate training data queue for each game is prepared.

During the *Update* phase, the DNN is trained using mini-batch stochastic gradient descent with a batch size of $N_{\text{batch}}$ for $N_{\text{epochs}}$ epochs. The optimizer uses weight decay. The loss function $l$ combines the mean squared error between the predicted value $v$ and the winner's disc color $c_{\text{win}}$, and the cross-entropy loss between the search probabilities $\boldsymbol{\pi}$ and the predicted move probabilities $\boldsymbol{p}$. The loss function is defined in Eq. (11):

$$l = (c_{\text{win}} - v)^2 - \boldsymbol{\pi}^{\text{T}} \log \boldsymbol{p}. \tag{11}$$

To train multiple games simultaneously, mini-batches are generated from the respective training data queue of each game. During the *Update* phase, mini-batches are sampled from these individual queues and used to update the DNN. For example, when an agent simultaneously trains Connect 4, Gomoku, and Othello, one mini-batch from the training data queues of Connect 4, Gomoku, and Othello is sequentially used to update the network.

## PARAMETERS

The hyperparameters for AlphaViT, AlphaViD, AlphaVDA, and AlphaZero are listed in Table A1. For training, the AdamW optimizer in PyTorch is used, with all parameters set to their default values except for the learning rate. All other parameters for AlphaZero were consistent with the previous implementation (*Fujita, 2022*). The hyperparameters of the other models were carefully hand-tuned to optimize their performance.

Table A2 lists the game-specific hyperparameters for AlphaViT, AlphaViD, AlphaVDA, and AlphaZero. The number of MCTS simulations ($N_{\text{sim}}$) ranges from 200 to 400, depending on the game and board size. The number of self-play games per iteration is set to 30 for Connect 4 variants and 10 for Gomoku and Othello variants. The opening phase ($T_{\text{opening}}$) specifies the number of initial moves using softmax decision-making with a temperature parameter ($\tau$) that is adjusted based on the game and board size.

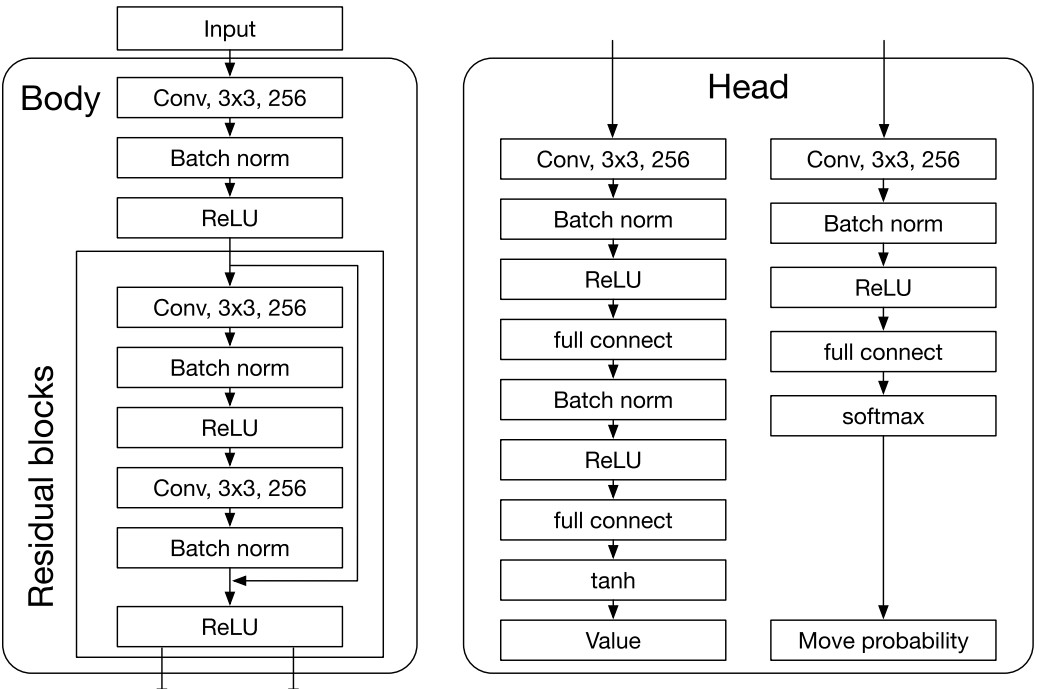

**Figure A1** DNN of AlphaZero.               

**Table A1** Hyperparameters of AlphaViT, AlphaViD, AlphaVDA, and AlphaZero.

| Parameter | AlphaViT | AlphaViD | AlphaVDA | AlphaZero |
|---|---|---|---|---|
| $C_{puct}$ | 1.25 | 1.25 | 1.25 | 1.25 |
| $\varepsilon$ | 0.2 | 0.2 | 0.2 | 0.2 |
| $T$ | 1 | 1 | 1 | 1 |
| $N_{queue}$ | 100,000 | 100,000 | 100,000 | 100,000 |
| $N_{epoch}$ | 1 | 1 | 1 | 1 |
| Optimizer | AdamW | AdamW | AdamW | AdamW |
| Batch size | 1,024 | 1,024 | 1,024 | 1,024 |
| Learning rate | 0.0001 | 0.0001 | 0.0001 | 0.0001 |
| Weight decay | 0.01 | 0.01 | 0.01 | 0.01 |
| Patch size | $5 \times 5$ | $5 \times 5$ | $5 \times 5$ | – |
| Patch stride | 1 | 1 | 1 | – |
| Embedding size of encoder | 512 | 512 | 512 | – |
| Encoder feedforward dimension | 1,024 | 1,024 | 1,024 | – |
| Number of encoder heads | 8 | 8 | 8 | – |
| Size of positional embeddings | $512 \times 256$ | $512 \times 256$ | $512 \times 256$ | – |
| Number of decoder layers | – | 1 | 1 | – |
| Embedding size of decoder | – | 512 | 512 | – |
| Decoder feedforward dimension | – | 1,024 | 1,024 | – |
| Number of decoder heads | – | 8 | 8 | – |

| Parameter | AlphaViT | AlphaViD | AlphaVDA | AlphaZero |
|---|---|---|---|---|
| Size of action embeddings | – | – | 256 | – |
| Dropout rate | 0.1 | 0.1 | 0.1 | – |
| Number of residual blocks | – | – | – | 6 |
| Kernel size | – | – | – | 3 |
| Number of filters | – | – | – | 256 |

**Table A2** Game-specific hyperparameters for AlphaViT, AlphaViD, AlphaVDA, and AlphaZero.

| | Connect 4 | Connect 4 5 × 4 | Gomoku | Gomoku 6 × 6 | Othello | Othello 6 × 6 |
|---|---|---|---|---|---|---|
| Number of simulations | 200 | 200 | 400 | 200 | 400 | 200 |
| Number of self-play | 30 | 30 | 10 | 10 | 10 | 10 |
| $T_{opening}$ | 4 | 4 | 8 | 6 | 8 | 6 |
| $\tau$ | 100 | 100 | 40 | 20 | 80 | 40 |

# ACKNOWLEDGEMENTS

The author gratefully acknowledges the assistance of large language models (ChatGPT, Google Gemini, Mistral Large 2, and Qwen2.5) and AI tools (Grammarly and Paperpal) in improving the grammar and style of this manuscript.

## Funding

The authors received no funding for this work.

## Competing Interests

The authors declare that they have no competing interests.

## Author Contributions

- Kazuhisa Fujita conceived and designed the experiments, performed the experiments, analyzed the data, performed the computation work, prepared figures and/or tables, authored or reviewed drafts of the article, and approved the final draft.

## Data Availability

The raw data and code are available on GitHub and Zenodo:

- https://github.com/KazuhisaFujita/AlphaViT.

- Kazuhisa Fujita. (2025). KazuhisaFujita/AlphaViT: New release (release). Zenodo. https://doi.org/10.5281/zenodo.17204839.

The trained weights of the models are available on Hugging Face: https://huggingface.co/kazufujita/AlphaViT.

https://doi.org/10.57967/hf/6826.

## Supplemental Information

Supplemental information for this article can be found online at http://dx.doi.org/10.7717/peerj-cs.3403#supplemental-information.

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
