# Peer review of "AlphaViT: a flexible game-playing AI for multiple games and variable board sizes"

_PeerJ Computer Science, doi:10.7717/peerj-cs.3403_

## Round 0.1 · original submission · Major Revisions

· Academic Editor

Major Revisions

**Language Note:** The review process has identified that the English language must be improved. PeerJ can provide language editing services - please contact us at [email protected] for pricing (be sure to provide your manuscript number and title). Alternatively, you should make your own arrangements to improve the language quality and provide details in your response letter. – PeerJ Staff

Reviewer 1 ·

Basic reporting

The author uses clear, unambiguous, professional English with only a few exceptions:
In line 412 the year of one citation is missing and replaced with a "?".
In line 426 the word "Figure" is abbreviated by Fig. while in the rest of the text the longer version "Figure" is used.
In Formula (2), using the same symbol twice is uncommon if you want to replace or update a value.

The article mentions much relevant literature in the related work section but fails to reference other previous works that applied the transformer architecture in the AlphaZero framework. To name a few, there is the work on "Grandmaster-Level Chess Without Search by Ruoss et al. (DeepMind), "Representation Matters for Mastering Chess: Improved Feature Representation in AlphaZero Outperforms Switching to Transformers" by Czech et al. and Mastering Chess with a Transformer Model by Monroe et al.

The article structure generally conforms to PeerJ standards, but the discussion and conclusion are currently one section, which could be divided into two sections.

The figures are generally relevant and of good quality. However, Figure (1) could be downscaled because its letter size is much higher than the rest of the text.
Additionally, some figures could be improved by giving the main message of its content in the description.
For example, the readability of Table 3 could be improved by marking the highest Elo values in bold and conveying the main message of the table in the table description.

Raw data here in the form of the source code and trained models are provided via GitHub and Hugging Face.

The main motivation of this paper is that A0's standard DNN is restricted to a single input size due to its use of MLPs, as stated in the introduction.
However, it is stated three times: line 35 (introduction) line 76, 93 (related work), line 110 (ALPHAVIT, ALPHAVID, AND ALPHAVDA)
It is a bit redundant to state it in the related work section repeatedly.

Experimental design

Methods and hyperparameter information are given to replicate the result. However, information about the computing infrastructure is missing.
The evaluation criteria, here the Elo metric, and background about AlphaZero and PUCT search sufficiently described.

There is no research question mentioned explicitly. However, the reader may assume it is to evaluate the new transformer architecture design in terms of performance, transferability to other games, and board sizes.

Multiple experiments are provided to answer questions about its transferability and comparison to standard AlphaZero.

Regarding the experimental setup (Table A1), it does not make sense to me why the A0 setup uses a different optimizer (here SGD vs AdamW), weight decay, and learning rate compared to the different transformer architectures. This may distort the actual results as to compare only the architecture changes alone.
Also, it is unclear to me what epsilon is referring to in this context.

Additionally, the number of parameters is given in Table 1, where AlphaZero has the lowest number of parameters and still performs best.
The number of floating point operations (FLOPS) and the inference time per model are missing here.
It is, e.g., unfair to expect the same performance under the same number of MCTS iterations when a model is twice as fast as a different model.
Moreover, it is mentioned in Figure 3 that "interpolation" is used for the embeddings, but it is missing what kind of interpolation is actually used here.

In line 176, it is described to use Sigmoid for the policy head output. This is unusual. Why did you not use Softmax here to have a distribution over all action within 0-1 that sums up to 1.0?

Validity of the findings

The underlying data of the experiments could be replicated, given the provided models.
Additionally, the experimental results support the stated conclusion, and the implicitly stated research question is answered.
The presented architecture designs AlphaViT, AlphaViD, and AlphaVDA are well evaluated but lack some important comparisons, as described in the following.

The author mentions using transformers to address the limitation of A0 (i.e., limitation to a single game and board size for a single DNN).
However, the author also correctly cites the work by Wu (2019) that tackles the same problem in a much simpler way by using a pure CNN architecture and avoiding MLPs.
Unfortunately, this paper does not compare against this model type and, therefore, fails to show a meaningful impact.
The proposed architecture design is first more complicated, presumably more costly than the standard CNN, and fails to achieve the same or better level of performance.
In contrast, it was shown in the work "Representation Matters for Mastering Chess: Improved Feature Representation in AlphaZero Outperforms Switching to Transformers" by Czech et al. and Mastering Chess with a Transformer Model by Monroe et al. that transformers or attention layers can help to boost performance.
In the current state, it is unclear to me what the main benefits and advantages are of using the new proposed methods compared to the CNN by Wu (2019).

Additional comments

Strengths
+ Overall, it has a good structure and is easy to follow.
+ Evaluation on a broad set but relatively simple environments.
+ Good understandable language.

Weaknesses
- Claims to handle multiple games and board sizes could also be done via standard CNNs.
- A comparison to standard CNN for multiple board sizes is lacking
- Performance of the final models is inferior compared to standard A0, models are much more complicated to design and presumably slower.
- Different sets of hyperparameters for training the A0 model compared to the transformer models.

Cite this review as

Reviewer 2 ·

Basic reporting

The paper is overall-well written, self-contained, and well-organized. The languages used are clear and accurate.

Experimental design

The performances are evaluated with Elo ratings, which are common in the area of game AI. Experiments compare with AlphaZero on several games. However, the method is evaluated on Go games, which is expected to have a larger scale the games evaluated.

Validity of the findings

Given that AlphaZero has already achieved good results across multiple games, the novelty of changing the neural architecture to vision transformer is limited. Also, it is unclear how scalable the method is, especially to larger-scale games, as transformers often require extensive computations.

Additional comments

Evaluating the method on larger scale games will strengthen the contributions of this paper. If experiments are not possible, the authors should at least discuss the scalability, especially when comparing with AlphaZero.

Cite this review as

Reviewer 3 ·

Basic reporting

The paper is generally well written. It presents the research problem clearly, and states the methods aimed to improve the current state of the art.

The author notes that the current board game top performer algorithm AlphaZero works best on the same table size as used during training. This happens due to the trailing inflexible MLP block that can be replaced with more versatile vision transformers.

The paper explains how it can be done and reports experimental data.

Experimental design

The design is straightforward. The author relies on a variety of board games, including Connect4, Gomoku, and Othello. The author, unfortunately, does not provide any rationale for this selection. To me, all these games are roughly of Checkers complexity, which means they seemingly can be played at superhuman level with traditional AI methods, and thus perhaps a bar too low for AlphaZero and related approaches.

Validity of the findings

I found the obtained results somewhat underwhelming, not entirely reliable, and reported in a way that could be improved.

First, it's hard to see who is better in each column of Table 3. It would be great, for example, to use bold font for the top 3 competitors in each case. Next, it seems that the improvement obtained turns out to be marginal, as AlphaZero seems to be still going strong despite being technically "inflexible" and having much smaller number of parameters. The author notes that the proposed approach can still be fine-tuned. Possibly so, but what about fine-tuning AlphaZero at least in terms of sheer model size? I am also unsure why MCTS scores relatively low -- I expected it to be able to compete on par with other approaches at least in simpler games like Connect4. It might be that MCTS here isn't really optimized to the extent possible, and doesn't have a fair chance to compete.

In terms of text, sec. 5.1 and 5.2 are poor: they merely list what is in the table 3 without providing much additional insight. By reading them, the reader won't gain anything. Sec. 5.2 even has long lines of numbers that are supposed to be... read? The author provides several pages of visuals and still believes they have to be explained in details. I'd focus on the meaning of the results and cut much of the existing text.

Finally, from Fig. 5 I'd question whether the reported data can be trusted. I believe it is reported accurately, but is it statistically significant? Elo ratings in some of the charts jump from iteration to iteration, and is not clear whether the point of time where algorithm X is better than Y is representative.

Additional comments

Summing up, I think the author tested an interesting idea and reported obtained findings accurately. However, the selection of games is unclear, the skill level of baseline models (whether they were operating on their state of the art level) is unclear, and final ranking table is not 100% reliable, and the obtained results are not as clear-cut as one might have hoped.

Cite this review as

·

Basic reporting

1. The article is generally clearly written. The introduction and motivation for the work are clear, well-structured and well-articulated. Contributions are clear: replace residual blocks in DNN of AlphaZero architecture with Vision Transformers, allowing the use of a single DNN with inputs of varying sizes, trainable to play several games at the same time.

2. The structure of information presented could be improved in certain areas. For example, it's unclear in Figure 2 where the game, value and pass embeddings come from, what they mean and what they are used for; this is explained later in section 3.1, the content could be better structured to allow natural understanding.

3. Literature on AlphaZero-style architectures is not very recent (before 2020), in a domain that's received a lot of attention and advances. An update should be done to more accurately reflect other follow-up algorithms and improvements. Some references are also missing, e.g. "(Silver et al., 2016,?, 2018)". MCTS should be cited on first mention.

4. Figures are quite large, the text in the diagram doesn't need to be larger than the paper's regular text (and can even be smaller, in line with the caption size). This would allow figures to generally be smaller and better integrated in the text, rather than taking up whole pages by themselves and making the reader scroll up and down the paper to find the figures referenced. Figures should generally be closer to the part of the text they're referenced in so it's as easy as possible to read the text, see the figure, and gain understanding of the concepts explained. Figure captions can be more concise and describe strictly the visual portrayed.

5. Several concepts are not described or not clear, and information is lacking especially in experimental setup documentation:
- Unclear why 'board states' is plural in Figure 1, what does this input actually look like? Is it more than one state that is passed at a time to the DNN?
- Unclear why the pass embedding used in AlphaViT is no longer used in the other 2 variants.
- No details on the training are given (how long the algorithms are trained for, hardware used etc.)
- It is unclear how the number of encoder layers were chosen for each algorithm tested.
- "were evaluated at iterations 1, 2, 4, 6, 8, 10, 20, 40, 60, 310 80, 100, 200, 300, 400, 500, 600, 700, 800, 900, 1000, 1100, 1200, 1300, 1400, 1500, 1600, 1700, 1800, 311 1900, 2000, 2100, 2200, 2300, 2400, 2500, 2600, 2700, 2800, 2900, and 3000" - this is a very long sequence of numbers, surely it could be summarised. Similar for the one just below. It's not clear what an "iteration" is, or why the numbers are different for small and large board variants of the games.
- It's not clear if sections 5.2 onwards are rerun experiments and evaluation tournaments than what is presented in 5.1
- It's not clear why AlphaViT L4 is not considered to have a deep encoder, is the boundary set at 5 layers?

6. Table 3 could be better organised to parse all the different numbers and better formatted. The best Elo rating in each column could be highlighted, for example. Some value could also be computed to determine the best agent overall (or overall performance). The baseline (AlphaZero) could also be highlighted; other Elo ratings could be displayed in terms of this rather than raw numbers for clearer presentation too (e.g. -1332.2 for Random and -234 for AlphaViT L4 LB in Connect4). Some colour-coding could also help, as would separating sections of the agents (e.g. grouping all AlphaViT, then AlphaViD, then AlphaVDA, then baseline, then other agents included, with spacing in-between). In its current format this table is really not digestable.

7. Some indication of human performance in all of these games could also be given. In the discussion there are mentions of 'strong' performance, but what does this really mean? Where do you draw the line for 'strong' or 'weak' performance? Unclear analysis. This is also rather shallow, with some conclusions drawn without a strong basis. Most of the discussion summarises results in the table rather than trying to interpret it.

8. The 'Multi' agents are tested on the same games seen during training and report better performance, is this not explainable through overfitting? How do they perform on different games?

Experimental design

1. The research carried out is original and within the scope of the journal. The research question is well defined, relevant & meaningful. It is stated how research fills an identified knowledge gap.

2. As detailed above, there is not enough information given on methodology to allow result replication, and several choices raise questions on the technical rigour of the experiments as well as the validity of the results.

Validity of the findings

Enough data is provided, but the results are not clear and validity cannot easily be assessed given lack of detail in methodology description. Conclusions are clear, but several are drawn without a strong basis. Limitations could be better discussed.

Additional comments

1. "AlphaViD, and AlphaVDA" - acronyms not defined before use in abstract.
2. "for calculating 129 value and move probabilities, as shown in 3" - unclear what '3' is.
3. "MLP (NLP_p)." - I guess that's MLP_p
4. Equations should be referred to in text appropriately.
5. "This study evaluates the performance of AlphaViT and AlphaViD across six games:" - 3 games are listed, each with 2 board sizes. This is more precisely recapped in Section 5.
6. Figure 4 has a lot of overlapping lines, making it hard to understand.

---

## Round 0.2 · Minor Revisions

· Academic Editor

Minor Revisions

Reviewer 1 ·

Basic reporting

The author rewrote major parts of the paper, and it is written in professional English.

The paper now also refers to additional related work, which was missing previously.

They also fixed a missing reference pointer.

Experimental design

I thank the author for adding new experiments and rewriting parts of the paper.

The author addressed some main critique points of the paper.

Mainly, the new experiment against the work by Wu et al. was implemented and conducted.

The results, however, were unexpected:
The author compared Wu et al.'s approach using a Res16 (ResNet with 16 residual blocks) against the standard ResNet Res6 with only 6 residual blocks.

Despite A0GP-Res16 having a much higher model capacity, it resulted in significantly lower performance, and the performance even decreased throughout training.

This seems very odd to me. I would expect the result of the network architecture in Go to translate to Connect4, Gomoku, and Othello, which are all much simpler games.

Moreover, the new experiment is only mentioned in the author's reply file. I'd suggest at least mentioning it in the supplements, if not in the main document.

Besides that, the different architectures now use the same optimizer, the same learning rate, and weight decay setting as detailed in Table A1. I approve of that.

Validity of the findings

The results of the new AlphaZero transformer models (AlphaViT, AlphaViD, AlphaVDA) offer more flexibility than the native standard AlphaZero as they support different board sizes.

However, the new models are often slightly worse than standard AlphaZero when it comes to performance.

I am also still a bit puzzled that the work of Wu et al, which introduces a generalized convolution network across board sizes, would not generalize to games outside of Go.

I suspect there might be something wrong when he reimplemented the global pooling block, but I may be wrong about that. It would be good to double-check the code against the source code at https://github.com/lightvector/KataGo

Cite this review as

Reviewer 3 ·

Basic reporting

Given this is the second round of review, I simply wanted to confirm that the author has revised the manuscript to address earlier comments. At least for my part, I think they are satisfactory, and possible further criticisms reflect personal preferences.

Experimental design

Experimental design is appropriate

Validity of the findings

The findings are reasonably validated

Cite this review as

---

## Round 0.3 · accepted · Accept

· Academic Editor

Accept

The authors have revised the comments from reviewer(s). Accept.

Reviewer 1 ·

Basic reporting

I greatly appreciate the author's effort to further improve this manuscript.
All my points have now been addressed.
I also noticed the details of all the figures and their good quality now.
In my opinion, this paper is now acceptable.

Experimental design

The author now provides detailed supplemental material on the AlphaZeroGP experiment, which is greatly appreciated.
The author also refers to this supplementary material in the main text.

Validity of the findings

The experiments appear to be valid from my perspective.

Additional comments

No additional comments.

Cite this review as